# LEARNING HJB VISCOSITY SOLUTIONS WITH PINNS FOR CONTINUOUS-TIME REINFORCEMENT LEARNING

## ABSTRACT

Despite recent advances in Reinforcement Learning (RL), the Markov Decision Processes are not always the best choice to model complex dynamical systems requiring interactions at high frequency. Being able to work with arbitrary time intervals, Continuous Time Reinforcement Learning (CTRL) is more suitable for those problems. Instead of the Bellman equation operating in discrete time, it is the Hamiltonian Jacobi Bellman (HJB) equation that describes value function evolution in CTRL. Even though the value function is a solution of the HJB equation, it may not be its unique solution. To distinguish the value function from other solutions, it is important to look for the viscosity solutions of the HJB equation. The viscosity solutions constitute a special class of solutions that possess uniqueness and stability properties. This paper proposes a novel approach to approximate the value function by training a Physics Informed Neural Network (PINN) through a specific $\epsilon$-*scheduling* iterative process constraining the PINN to converge towards the viscosity solution and shows experimental results with classical control tasks.

## 1 INTRODUCTION

Reinforcement learning (RL) is getting more and more attention. Most of state-of-the-art RL methods are designed to work with Markov Decision Processes (MDPs) and, in particular, rely on a discrete time assumption. Even though this assumption is not that restrictive in some tasks like games, it is no longer valid in complex problems such as driving cars, finance trading or controling the dynamical system. Such problems are usually described by a dynamical system where discrete time RL (DTRL) often struggles to provide accurate control within short time intervals due to several reasons (Doya, 2000; Wang et al., 2020; Mukherjee & Liu, 2023). As DTRL algorithms operate in regular discrete time steps, it becomes challenging to capture the nuances of continuous state dynamics within small intervals. Even when the timestep is set to be small, DTRL may fail to learn the optimal policy as exploration cannot be done efficiently.

Continuous Time Reinforcement Learning (CTRL) derives from the optimal control theory and thus provides a promising theoretical framework to tackle aforementioned shortcomings of DTRL. Moreover, CTRL is agnostic on the discretization of time and thus can provide a good control at any chosen frequency without the need to retrain the policy. Similar to DTRL, one way to find an optimal policy is to compute it using the value function. Thus, the key ingredient in CTRL is the Hamilton-Jacobi-Bellman (HJB) equation (Doya, 2000; Munos, 2000), a Partial Differential Equation (PDE), which is a continuous-time counterpart of the Bellman equation. This PDE describes the evolution of the value function with respect to a state of the system, which in turn evolves continuously with time. In principle, the value function is a solution of the HJB equation. However, it is only a necessary condition and not sufficient. The HJB equation may have multiple solutions in a class of continuous functions (Munos, 2000) making it a challenging task to find the value function. Distinguishing the value function from the other solutions of the HJB equation relies on the search for the *viscosity* solution that possesses uniqueness and stability properties.

However, there is no guarantee in general that the solution found by existing methods is the good solution, *i.e.* the viscosity solution. Moreover, even verifying if a given solution is a viscosity solution is a hard task let alone finding them. The existing methods for solving the HJB equation are limited and mainly applicable to specific cases such as Linear Quadratic Regulator (LQR) problems or

variational problems (Fleming & Soner, 2006). There exist approaches that can be applied in general case, such as Finite Difference (FD), Finite Element Methods (FEM) (Grossmann et al., 2007), and dynamic programming methods (Munos, 2000) that transform a CTRL problem back to a DTRL problem thanks to the discretization. However, their effectiveness is hindered by the exponentially growing algorithmic complexity with respect to the state space dimensionality. In addition, they are also affected by discretization error.

Conversely, neural networks (NNs) are emerging as PDE solvers that can cope with the curse of dimensionality (Raissi et al., 2019). Although Neural Networks have shown a great potential in various domains (Lu et al., 2021; Karniadakis et al., 2021), applying them to solve the HJB equation requires caution because of the aforementioned non-uniqueness issue.

In this paper, we revisit CTRL approaches and analyze how the latest advances in deep learning can be applied and adjusted to solve the HJB equation in the viscosity sense. The main contribution of this paper is to provide the NN based framework to find the viscosity solutions of the general HJB equation. To the best of our knowledge it is the first attempt to do it. Further, we focus on the case of deterministic environments with known dynamics. The extension to stochastic environments and unknown dynamics is possible (see Wang et al. (2020); Çağatay Yıldız et al. (2021)), but left for the future work. To achieve viscosity, our approach is to solve sequentially a series of PDE equations so that the solution of the final one is the value function. We use NN solvers for those PDEs and we propose several ways of building the sequence of those equations. Thus, this work can be interesting for the optimal control community as we show how to use the neural networks to get the viscosity solutions and for the reinforcement learning community as our work can be used as a basis for Model Based Continuous Time Reinforcement Learning.

This work is divided in several parts. We start with the analysis of the existing literature in Section 2. Then, we carefully introduce the definitions and notations related to CTRL in Section 3.1, describe the HJB equation in Section 3.2 and define viscosity solutions in Section 3.3. Then, different ways of integrating neural networks in the process of solving the HJB equation are discussed in Section 4. Finally, we demonstrate performances of our approach from Section 4 on the inverted pendulum, a classical use case from RL, and compare with DTRL methods on some other classical control tasks in Section 5.

## 2 RELATED WORKS

Among the first papers to study CTRL are Doya (2000); Munos (2000); Coulom (2002). Those works introduced the HJB equation as a key equation for finding the optimal policy. In Doya (2000), different discretization schemes and algorithms are analysed for computing the value function, including continuous TD($\lambda$) and continuous Actor-Critic. In his Ph.D. (Coulom, 2002), Coulom studied applicability of Doya's methods (Doya, 2000) to a large class of control problems. Munos (Munos, 2000) tackled CTRL by the study of viscosity solutions and their properties. He demonstrated the challenges of solving the HJB equation such as the non-uniqueness of solutions and inequality boundary conditions. In addition, convergent numerical schemes based on dynamic programming were derived to approximate the value function. One of the problems of numerical schemes is the curse of dimensionality. To mitigate this problem, sparse grids (Kang & Wilcox, 2016) can be used instead of naive uniform grid. In this article, we take the formalism proposed in Munos (2000), but we consider neural network based approaches to find viscosity solutions, while Munos (2000) considers tabular algorithms.

NNs can be applied for solving the HJB equation (Munos et al., 1999; Liu et al., 2014; Cheng et al., 2007; Tassa & Erez, 2007; Lutter et al., 2020; Han et al., 2018; Adhyaru et al., 2011). It was first demonstrated in Munos et al. (1999). In Tassa & Erez (2007), the training is regularized to avoid finding "bad" solutions. Moreover, the same work raised the problem of falling in a local minimum of the squared HJB residual, and some solutions to avoid it were proposed. Compared to previous papers, Liu et al. (2014); Adhyaru et al. (2011) have emphasized the robustness of the resulting controller and the stability of the proposed algorithm. Adapting the former methods for more practical cases such as the inverted pendulum and the cartpole was done in Lutter et al. (2020). We consider similar approaches as Tassa & Erez (2007); Lutter et al. (2020), but unlike them we use the formalism proposed in Munos (2000) to develop an approach that can converge to viscosity solutions.

Several extensions to the classical HJB equation were also introduced such as HJB for an explorative reward function (Wang et al., 2020), the soft HJB equation with maximum entropy regularization (Kim & Yang, 2020; Halperin, 2021) and the distributional HJB equation (Wiltzer et al., 2022). Those works extend the existing theory to other definitions of the value function, however experiments are conducted on a limited set of simple problems. Futhermore, it is also possible to extend the HJB equation to continuous-time partially observable Markov decision processes (CTPOMDPs). In (Alt et al., 2020), they proposed a formalism to describe CTPOMDPs, including the CTPOMDP HJB equation. Kim et al. (2021) tackles the problem of CTRL by adapting the well-known DQN algorithm to this framework. A definition for the Q-function in the continuous case is given and the "HJB equation for the Q-function" is derived, which results in a DQN-like algorithm for the semi-discrete time setting. However, this approach is limited to Lipschitz continuous control. The HJB equation was used to improve DTRL algorithms like PPO in Mukherjee & Liu (2023) and it resulted in a significant improvement on MuJoCo, proving that HJB loss is better adapted for learning value functions of dynamical systems.

There were some attempts to propose alternative ways for solving the HJB equation. An other NN approach to approximate the value function based on Pontryagin's maximum principle has been proposed (Nakamura-Zimmerer et al., 2021). Conversely, Darbon et al. (2023) considers some special neural network architectures that can work with min-plus algebra, though this approach is suitable only for some optimal control problems.

Another line of research exploits the continuous time formulation to do more accurate Monte Carlo estimations of value function (Lutter et al., 2021b;a; Çağatay Yıldız et al., 2021) rather than using the HJB equation. Lutter et al. (2021b) adapted the fitted value iteration algorithm to the CTRL problem. In Çağatay Yıldız et al. (2021), a model-based algorithm was introduced, which aims at solving the problem in an actor-critic manner. Bayesian neural ODEs are used in order to learn the dynamics of the system.

## 3 HAMILTON-JACOBI-BELLMAN EQUATION

### 3.1 FORMALISM FOR REINFORCEMENT LEARNING IN THE CONTINUOUS CASE

In this work, we consider the optimal control problem for infinite-horizon deterministic dynamical systems. The dynamics are known and reward signal is giving at each control step. The extension to unknown dynamics and rewards is left for future works.

In what follows, we denote with $O \subseteq \mathbb{R}^d$, the open set of *controlable* states of our system, and then the *admissible* control $u \in U$ keeps the state trajectory inside the domain $O$, where $U \subset \mathbb{R}^m$ corresponds to the action/control space. We assume that $U$ is bounded. We use $g \in C(O)$ to denote that $g$ is a continuous function on $O$, while $g \in C^1(O)$ that $g$ is a continuously differentiable function on $O$.

The main difference between the continuous time and the discrete time cases is that transitions depend on time $t$, which is a continuous variable, generating trajectories of states continuously in time. More formally, in the continuous case the states do not form a sequence $\{x_t\}_{t=0}^{\infty}$ but a trajectory $x : \mathbb{R}_+ \to \bar{O} \subset \mathbb{R}^d$. Similarly, the actions are defined for any $t$: $u : \mathbb{R}_+ \to U$. The dynamics of the environment (the transition function) is defined through the following ordinary differential equation (ODE):

$$\frac{dx(t)}{dt} = f(x(t), u(t)) \quad \text{or} \quad x(t') = x(t) + \int_t^{t'} f(x(s), u(s)) \, ds, \ t' > t \tag{1}$$

where $f : \bar{O} \times U \to \mathbb{R}^d$ is called the state dynamics function. To compare, $x(t)$ is only defined at times $\{0, dt, 2dt, \dots\}$ in discrete time case, where $dt$ is a time step. While the state $x$ is inside the control domain $O$ (*i.e.* $x(t) \in \bar{O}$), the reward $r : \bar{O} \times U \to \mathbb{R}$ is received.

Without loss of generality, we focus on the problem of optimal contol under state constraints in this paper, *i.e.* $x(t) \in O$ for any $t > 0$ and we define the further notations related to the problem from this standpoint. Note that it is a common requirement in the dynamical systems. For example, we may want to limit the maximal angular speed in the inverted pendulum to mitigate the risk of wearing off or breaking the mechanism. The interested reader can refer to Appendix A.1.4 to learn more about other possible cases.

Given the initial state $x(0) = x_0$, we define the cumulative discounted reward for a given control function $u(t)$ as follows

$$J(x_0; u(t)) = \int_0^\infty \gamma^t r(x(t), u(t)) dt, \tag{2}$$

where $\gamma \in [0, 1)$ is a discount factor. $J$ is called the reinforcement functional. Then, the value function is defined as:

$$V(x) = \sup_{u(t) \in U_x} J(x; u(t)), \tag{3}$$

where $U_x = \{u(t) | x(t) \in O, \forall t > 0 \text{ and } x(0) = x\}$, *i.e.* control that keeps the state of the system inside the domain $O$. Our goal is to find an optimal policy $\pi : \bar{O} \to U$, such that $u^*(t) = \pi(x^*(t))$ for any $t$, where $u^*(t)$ and $x^*(t)$ are control and state of the optimal trajectory respectively, such that $V(x) = J(x; u^*(t))$.

### 3.2 HAMILTON-JACOBI-BELLMAN EQUATION

Let $H(x, W, \nabla_x W) = -W(x) \ln \gamma - \sup_{u \in \mathcal{U}} [\nabla_x W(x)^T f(x, u) + r(x, u)]$, then the HJB equation can be expressed as

$$H(x, W, \nabla_x W) = 0. \tag{4}$$

We can prove that the value function defined in equation 3 satisfies the following result (see Fleming & Soner (2006)):

**Theorem 3.1** *(Hamilton-Jacobi-Bellman). If the value function $V$ is differentiable at $x$, then it should satisfy the Hamilton-Jacobi-Bellman (HJB) equation.*

If the value function $V(x)$ is known, then we can define a feed-back control policy $\pi : \bar{O} \to U$ such that $\pi(x(t)) = u^*(t)$ by setting:

$$\pi(x) \in \underset{u \in U}{\operatorname{argsup}} \left\{ \nabla_x V(x)^T f(x, u) + r(x, u) \right\} \tag{5}$$

When $O \neq \mathbb{R}^d$, the control in state constrained optimal control problems should also satisfy $f(x, u^*(x))^T \eta(x) \leq 0$ for any $x \in \partial O$, where $\eta(x)$ the external normal vector at point $x \in \partial O$. The optimal policy is not known a priori and thus it is hard to verify this constraint. In Fleming & Soner (2006); Soner (1986), it was shown that it can be reformulated as:

$$- H(x, W, \nabla_x W + \alpha \eta(x)) \leq 0 \quad \forall \alpha \leq 0, x \in \partial O. \tag{6}$$

From equation 5, it is crucial to find an efficient way to compute $V$ to get the optimal control $u^*$. In DTRL, the Bellman equation is traditionally used to find $V$. The HJB equation can be seen as a continuous-time analog of the Bellman equation. However, solving the HJB equation involves several challenges (see Munos (2000)). First, the value function $V$ is often non-smooth, and only continuous on $O$. Second, the HJB equation may have multiple generalized continuous solutions. Third, it is common that the HJB equation (equation 4) has to be solved under inequality boundary conditions. To address those points, we introduce the viscosity property.

### 3.3 VISCOSITY

Here, we present the crucial and yet complex notion of viscosity (refer to Appendix A.2 for an intuition behind).

**Definition 3.1 (Viscosity solution)**

- *$W \in C(O)$ is a viscosity subsolution of the HJB equation in $O$ if $\forall \psi \in C^1(O)$ and $\forall x \in O$ local maximum of $W - \psi$ such that $W(x) = \psi(x)$, we have:*

$$H(x, \psi(x), \nabla_x \psi(x)) \leq 0$$

- *$W \in C(O)$ is a viscosity supersolution of the HJB equation in $O$ if $\forall \psi \in C^1(O)$ and $\forall x \in O$ local minimum of $W - \psi$ such that $W(x) = \psi(x)$, we have:*

$$H(x, \psi(x), \nabla_x \psi(x)) \geq 0$$

- *If $W \in C(O)$ is a viscosity subsolution and a supersolution then it is a viscosity solution.*

Viscosity allows us to separate "good" solutions of the HJB equation from "bad" ones. Viscosity solutions were first introduced in Crandall & Lions (1983) and they are proven to be unique for multiple types of PDEs. Under some additional assumptions, which includes continuity of $f$ and $r$, one can prove that the value function is a *unique viscosity solution* for $O = \mathbb{R}^d$ (See Appendix A.1.3 for more details). The similar result exists for $O \subseteq \mathbb{R}^d$ and when the value function should satisfy the boundary condition equation 6.

The uniqueness of the value function as a viscosity solution holds for the large class of control problems with continuous dynamics that appear in practice, like classical control or MuJoCo (Todorov et al., 2012). Therefore, there is an interest of having the methods that are able to compute the viscosity solutions of the PDE equations. However, checking the conditions of Definition 3.1 is not feasible in practice. Instead, we use the next property of viscosity solutions.

**Lemma 3.1 (Stability)** *Let $W^\epsilon$ be a viscosity subsolution (resp. a super solution) of*
$$W^\epsilon(x) + F^\epsilon(x, W^\epsilon(x), \nabla_x W^\epsilon(x), \nabla_x^2 W^\epsilon(x)) = 0$$
*in $O$. Suppose that $F^\epsilon$ converges to $F$ uniformly on every compact subset of its domain, and $W^\epsilon$ converges to $W$ uniformly on compact subsets of $\bar{O}$. Then $W$ is a viscosity subsolution (resp. a supersolution) of the limiting equation.*

This lemma is proven in (Fleming & Soner, 2006). In our case, we are interested in equation:
$$H(x, W^\epsilon(x), \nabla_x W^\epsilon(x)) = \epsilon \operatorname{Tr} \nabla_x^2 W^\epsilon(x), \tag{7}$$
where the left hand side is the same as in equation 4, while the right hand side depends linearly on $\epsilon > 0$. Therefore, $F^\epsilon(x, W^\epsilon(x), \nabla_x W^\epsilon(x), \nabla_x^2 W^\epsilon(x)) = -\frac{1}{\ln \gamma} H(x, W^\epsilon(x), \nabla_x W^\epsilon(x)) - W^\epsilon(x) + \frac{\epsilon}{\ln \gamma} \operatorname{Tr} \nabla_{xx}^2 W^\epsilon(x)$ and $F(x, W(x), \nabla_x W(x), \nabla_x^2 W(x)) = -\frac{1}{\ln \gamma} H(x, W(x), \nabla_x W(x)) - W(x)$ in Lemma 3.1. In Fleming & Soner (2006), it is shown that equation 7 has a unique smooth solution $W^\epsilon(x)$, *i.e.* it admits a classical solution, which is a viscosity solution at the same time. Therefore, if $W^\epsilon(x)$ converges uniformly to $W(x)$ then $W(x)$ is a viscosity solution of the original HJB equation (equation 4).

## 4 How to Reach Viscosity

In what follows, we present two methods to find viscosity solutions of the HJB equation. First, we present the existing method based on dynamic programming. Then, we introduce a new neural approach. Further, we assume that the state dynamics $f(x, u)$ are known and the control space $U$ is discrete. The case of unknown $f(x, u)$ and continuous control space is left for future work.

### 4.1 Dynamic Programming

Several solvers exist such as Finite Difference method (FD) or Finite Element Method (FEM). These methods require the discretization of the domain (a grid for FD or a triangulation for FEM). The work of Munos (2000) establishes the connection between solving the HJB equation and the classical reinforcement learning framework by deriving an MDP from the discretization of the HJB equation, using either FD or FEM schemes (see Appendix A.3 for a short summary of the method). The strong point of this method is that there exist viscosity convergence guarantees (Munos, 2000). The weak point is that they are mesh-dependent, making it suffer from the curse of dimensionality. For example, in case of CartPole where $O \subseteq \mathbb{R}^4$, a naive approach that divides all axes uniformly in $N$ parts results in $N^4$ states to handle. Setting $N = 32$, which may not be sufficient to solve the problem, leads to $2^{20}$ states, which is already too many to process on a single device. In Section 5, we present the performance of the FEM based dynamic programming only on the inverted pendulum environment due to the aforementioned reasons. Despite many efforts, we were not able to make the algorithm based on FD work in our experiments, thus it is not considered.

### 4.2 Neural Solver

The dynamic programming approaches are able to find the solutions of the HJB equation that are intrinsically viscosity solutions. The latter is ensured by the choice of discretization step. In general, it

is hard to derive schemes that find the viscosity solutions directly. Conversely, checking the conditions of Definition 3.1 is not feasible in practice. Instead, we use the stability property of viscosity solutions (Lemma 3.1). That result lies at the heart of our approach. The idea is to solve a series of PDE equations starting from some $\epsilon_0$ and gradually decrease it so that the solution of the final PDE for $\epsilon_\infty = 0$ is the desired viscosity solution. Thus, the general framework to get viscosity solutions is the following: define the sequence of $\{\epsilon_n\}_{n=0}^\infty$, choose a PDE solver, then iteratively solve equation 7 so that $W^\epsilon(x)$ form a convergent sequence to $W(x)$ and output $W(x)$ as a final result. In what follows, we choose PINNs as a PDE solver and we define a few $\epsilon$-schedulers to generate $\{\epsilon_n\}$.

**PINNs** The idea of PINNs was proposed in Raissi et al. (2019) and applied to some simple PDEs like one-dimensional nonlinear Schrödinger equation, but not for optimal control. In PINNs framework, the neural network acts as a solution of the PDE that needs to be solved. Being randomly initialized, the neural network is gradually fit to satisfy the PDE and its corresponding boundary conditions with the help of optimization and automatic differentiation that allows to compute precise derivatives. For example, the solution of the equation $W_x' - W = 0$ for $W \in C^1(O = [0, 1])$ with the boundary condition $W(0) = 1$ can be found by minimizing the loss $\min_W\{\|W_x' - W\|_2^2 + \lambda\|W(0) - 1\|_2^2\}$. The first term of this loss is called a PDE loss and the second term a boundary loss, where $\lambda$ is a hyperparameter that weighs a boundary loss against a PDE loss. PINNs can be trained in a self-supervised manner as a dataset can be generated by simply drawing random samples from the domain $O$. Still, if the solution is known at some points of the domain, then the training can be augmented with a data-driven loss. Refer to Appendix A.4 for a more detailed introduction to PINNs.

Further, we denote $W^\epsilon(\cdot, \theta)$ and $W(\cdot, \theta)$ as neural networks that compute the solutions of equation 7 and equation 4 respectively with $\theta$ being its parameters. In PINNs-like mannner, we define losses corresponding to equation 7 and equation 6. Let us define $\mathcal{S}_O \sim \mathcal{U}(O, N_F)$, a sample of points drawn uniformly from $O$ of size $N_F$, and $\mathcal{S}_{\partial O} \sim \mathcal{U}(\partial O, N_B)$. Further, if $\epsilon$ and $\theta$ are indicated without an iteration number, then they correspond to the current iteration $n$. We have:

$$\mathcal{L}_O(\theta, \mathcal{S}_O) = \frac{1}{N_F}\sum_{i=1}^{N_F}\left(H(x_i, W^\epsilon(x_i, \theta), \nabla W^\epsilon(x_i, \theta)) - \epsilon\mathrm{Tr}(\nabla^2 W^\epsilon(x_i, \theta))\right)^2 \qquad x_i \in \mathcal{S}_O,$$

(8)

$$\mathcal{L}_{\partial O}(\theta, \mathcal{S}_{\partial O}) = \frac{1}{N_B}\sum_{i=1}^{N_B}\left([-H(x_i, W^\epsilon(x_i, \theta), \nabla W^\epsilon(x_i, \theta) + \alpha\eta(x_i))]^+\right)^2 \qquad x_i \in \mathcal{S}_{\partial O},$$

(9)

where $[f(x)]^+ = \max\{f(x), 0\}$. In addition to PDE-related and boundary-related losses, we introduce an MSE regularization loss that should encourage uniform convergence:

$$\mathcal{L}_R(\theta, \mathcal{S}_O) = \frac{1}{N_F}\sum_{i=1}^{N_F}\left(W^\epsilon(x_i, \theta) - W^{\epsilon_{n-1}}(x_i, \theta_{\epsilon_{n-1}})\right)^2 \quad x_i \in \mathcal{S}_O \qquad (10)$$

where $W^{\epsilon_{n-1}}(x, \theta_{\epsilon_{n-1}})$ is the best function obtained for $\epsilon_{n-1}$. The final loss is:

$$\mathcal{L}(\theta, \mathcal{S}_O, \mathcal{S}_{\partial O}) = \mathcal{L}_O(\theta, \mathcal{S}_O) + \lambda\mathcal{L}_{\partial O}(\theta, \mathcal{S}_{\partial O}) + \lambda_R\mathcal{L}_R(\theta, \mathcal{S}_O). \qquad (11)$$

**$\epsilon$-schedulers** To ensure the uniform convergence of equation 7 to equation 4, we need to define a sequence of $\{\epsilon_n\}_{n=0}^\infty$, such that $\epsilon_n \to 0$ and $\|\epsilon_n \mathrm{Tr}(\nabla_x^2 W^\epsilon(x))\| \to 0$ uniformly for all $x \in \bar{O}$ when $n \to \infty$. We propose three ways to define this sequence. The first is *non-adaptive scheduler*:

$$\epsilon_{n+1} = \begin{cases} \epsilon_n k_\epsilon & \text{if } n + 1 \equiv 0 \mod N_u \\ \epsilon_n & \text{otherwise} \end{cases} \qquad (12)$$

where $n$ corresponds to the current iteration number, $k_\epsilon \in (0, 1)$ a speed with which $\|\epsilon_n \mathrm{Tr}(\nabla_x^2 W^\epsilon(x))\|^2$ should decrease and $N_u$ the number of iterations between each update.

The second scheduler is called *adaptive scheduler*. Let $\delta(\epsilon, \theta) = \frac{1}{N_S}\sum_i\left\|\epsilon\mathrm{Tr}(\nabla_x^2 W^\epsilon(x_i, \theta))\right\|^2$. Given the first element of the sequence $\epsilon_0$, we get all the consecutive elements with the next update rule

$$\epsilon_{n+1} = \begin{cases} \frac{k_\epsilon\delta(\epsilon_n, \theta_{n-1})}{\delta(\epsilon_n, \theta_n)}\epsilon_n & \text{if } k_\epsilon\delta(\epsilon_n, \theta_{n-1}) \leq \delta(\epsilon_n, \theta_n), \\ & \text{and } \mathcal{L}(\theta_i) \geq \mathcal{L}(\theta_{i-1}) \,\forall i : n - n_\epsilon + 1 \leq i \leq n \\ \epsilon_n & \text{otherwise,} \end{cases} \qquad (13)$$

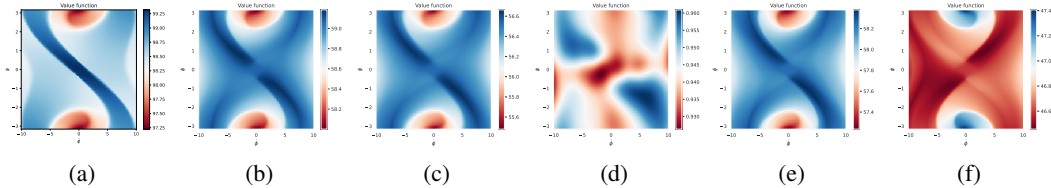

Figure 1: Value function of Pendulum for different $\epsilon$-schedulers: (a) ground truth, (b) hybrid, (c) non-adaptive, $k_\epsilon = 0.5$, $N_u = 10$ (d) non-adaptive, $k_\epsilon = 0.5$, $N_u = 75$ (e) adaptive, $\epsilon_0 = 1$, $k_\epsilon = 0.99$ (f) adaptive, $\epsilon_0 = 10^{-3}$, $k_\epsilon = 0.99$

where $n$ and $k_\epsilon$ have the same purpose as above. Moreover, $n_\epsilon$ serves as a number of iterations since $\theta_i$ does not improve the loss, *i.e.* it also specifies the minimum number of iterations with fixed $\epsilon$ to obtain $W^\epsilon(x, \theta) \approx W^\epsilon(x)$ that solves equation 7.

We also try a *hybrid scheduler*, mixing the two previous schedulers by starting with the non-adaptive scheduler for several $\epsilon$-updates and then using the adaptive scheduler until the end of the training. The nonadaptive scheduler serves as a "warm-start" at the beginning of the training allowing us to do more regular updates for large $\epsilon$ for which training is easier (see Appendix A.5.3). Then, it is better to use the adaptive scheduler for smaller $\epsilon$ to ensure the convergence $\delta(\epsilon, \theta) \to 0$. Finally, putting everything together gives Algorithm 1.

---

**Algorithm 1** $\epsilon$-HJBPINNs

---

Set $\epsilon = \epsilon_0$, $\theta = \theta_0$, initialize $W^\epsilon(\cdot, \theta)$
**for** iteration $n$ in $\{1, \ldots, \text{NB\_ITER}\}$ **do**
    Generate datasets $\mathcal{D}_O(N_D)$ and $\mathcal{D}_{\partial O}(N_D)$ of $N_D$ states uniformly sampled from $O$ and $\partial O$
    respectively
    **for** batches $\mathcal{S}_O \in \mathcal{U}(\mathcal{D}_O, N_S)$ and $\mathcal{S}_{\partial O} \in \mathcal{U}(\mathcal{D}_{\partial O}, N_S)$ **do**
        Update $\theta_n := \theta_n - \nu \nabla_\theta \mathcal{L}(\theta, \mathcal{S}_O, \mathcal{S}_{\partial O})$, where $\mathcal{L}(\theta, \mathcal{S}_O, \mathcal{S})$ is computed with equation 11
    **end for**
    Compute $\epsilon_n$ using equation 12 and equation 13
    Set $\epsilon = \epsilon_n$ and $\theta = \theta_n$
**end for**

---

## 5 EXPERIMENTAL RESULTS

**Analysis of $\epsilon$-Schedulers on Pendulum** In this section we evaluate the performance of Algorithm 1[1] for different $\epsilon$-schedulers on inverted pendulum environment. The state space consists of the angle $\phi$ and the angular speed $\dot{\phi}$. Here, we consider $O = [-\pi, \pi] \times [-20, 20]$ and $U = \{-2, 0, 2\}$.

Each training was executed for 300 iterations (NB\_ITER = 300). We tried 2 types of architectures, such as Multi-Layer Perceptron (MLP) and Fourier-Feature Network (FFN) (Yang et al., 2022) with different number of layers and neurons. We have obtained the best performances with FFN, consisting of 3 layers, where the first layer is of size $d \times 40$ (as recommended in the original article, where $d$ is the dimensionality of state space) and other layers are 100 neurons each. The best performing activation function is `tanh`. When working with PINNs, it is important to use smooth activation functions as using non-smooth activations like `relu` may cause the training to fail. Indeed, the PDE loss $\mathcal{L}_O$ requires computing second order derivatives, and even the third derivative during the backward propagation, but those derivatives do not exist for `relu`. We have also observed that the training is more stable if we *standardize* the output of the neural networks, *i.e.* $W^\epsilon(x_i, \theta)$ and $W^{\epsilon_{n-1}}(x_i, \theta_{\epsilon_{n-1}})$, across the samples in the batch $\mathcal{S}_O$ just before computing the regularization loss (equation 10). Indeed, the scale of the value function computed with the neural network is constantly changing during the training, therefore standardization helps to enforce uniform convergence without restraining the neural network to fit the solution of equation 7 for the current $\epsilon$.

---

[1]The placeholder for the repository link. The code will be available for the final version.

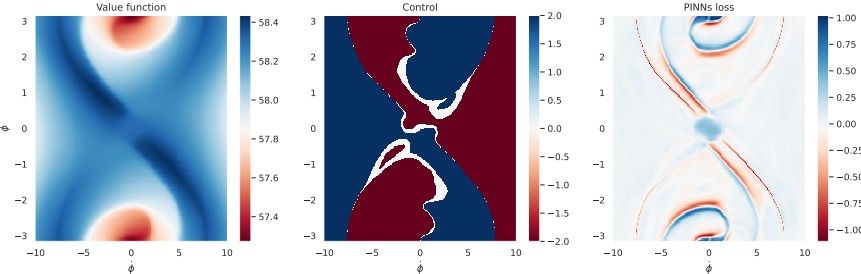

Figure 2: Value function (left), control (middle) and loss (right) maps obtained with PINNs after 300 iterations.

We have experimented with different $\epsilon$-schedulers from Section 4.2 and their hyperparameters. Some results are shown in Figure 1 and more can be found in Appendix A.5.2. To sum up, all three schedulers can learn the value function under some set of hyperparameters. However, one should be careful about the hyperparameters choice. The non-adaptive scheduler may fail if $\epsilon$ is updated too fast and the neural network is not able to adjust to a new $\epsilon$, but also it may fail if $\epsilon$ is updated too slow, as the neural network starts to overfit to the given $\epsilon$ (Figures 1c-1d). The adaptive scheduler performs well if it starts from high $\epsilon_0 = 1$, however it can be slow to converge, conversely starting from $\epsilon_0 = 0.001$ may fail for some seeds (Figures 1e-1f). The hybrid scheduler (Figure 1b) is the fastest to converge and the most robust with respect to different random initializations (We obtain the similar results across 8 different seeds). Next, we use the hybrid scheduler for our experiments, consult Appendix A.5.4 for the list of the best hyperparameters for the hybrid scheduler.

**PINNs Loss Map on Pendulum**    In Figure 2, we present the value function, control and loss maps that we have obtained after training. PINNs are able to find the general structure of the value function, but it is less precise when approximating non-smooth zones. This problem can be observed from Figure 2(right), where the PINNs loss (equation 8) has high values in the non-smooth zones from Figure 2(left). In previous works, it was observed that smooth activation functions may fail at grasping high frequency signals (Sitzmann et al., 2020). Thus, further research on how to choose the neural networks architecture for solving HJB is necessary.

**Comparison with DTRL and Dynamic Programming Algorithms**    In this section, we compare the performance of Algorithm 1 on different classical RL control tasks with well-studied DTRL algorithms such as PPO and A2C. We use a continuous-time adaptation of Inverted Pendulum, CartPole and Acrobot[2]. Those are challenging benchmarks similar to Lutter et al. (2020), where PINNs are also applied to solve the HJB equation.

*Pendulum ($dt = 0.001$)[3]* The state space consists of the angle $\phi$ and the angular speed $\dot{\phi}$. We consider $O = [-\pi, \pi] \times [-10, 10]$ (reducing the domain makes the comparison fairer with respect to exploration-based algorithms that do not compute value function on the whole domain) and $U = \{-2, 0, 2\}$.

*CartPole ($dt = 0.001$)* The state space consists of the pole angle $\phi$, the pole angular speed $\dot{\phi}$, the cart coordinate $y$ and its speed $\dot{y}$. We consider two problems: swing up the pole with $O = [-\pi, \pi] \times [-10, 10] \times [-5, -5] \times [-5, 5]$ and stabilizing the pole $O = [-\frac{1}{15}\pi, \frac{1}{15}\pi] \times [-10, 10] \times [-2.4, -2.4] \times [-5, 5]$ with $U = \{-3, 0, 3\}$ in both cases.

*Acrobot ($dt = 0.005$)* The state space consists of angles $\phi_1$ and $\phi_2$ and their corresponding angular speeds $\dot{\phi}_1$ and $\dot{\phi}_2$. The control is the torque applied to the extreme tip. We consider $O = [-\pi, \pi] \times [-\pi, \pi] \times [-12.57, 12.57] \times [-28.27, 28.27]$ and $U = \{-5, 0, 5\}$.

To test the algorithms, we perform 100 rollouts, each rollout being made of 5000 time steps. For the Pendulum task, we also show the performance of dynamic programming (DP) for the grid size

---

[2]We used the environments taken from `https://github.com/cagatayyildiz/oderl` and slightly modified them to define $O$ explicitly in each case

[3]In the gym inverted pendulum environment $dt$ is set to $1/20$

$N = 200$. A complete discussion of our experiments with dynamic programming can be found in A.5.1. To compare PINNs and DTRL agents, we used the same number of samples, however note that this setting is less advantageous for PINNs as it requires to sample uniformly across the whole domain to guarantee the viscosity, while DTRL agents can learn more from the trajectories that bring the highest outcome. For each algorithm, we take the best trained agent and we report its evalution mean and standard deviation over all rollouts in Table 1. Despite our best efforts, we could not make A2C learn on Acrobot with small $dt$, therefore this result is absent.

Even though, PINNs is performing worse than DP on Pendulum as it struggles to reach the optimal control in the area where the value function is non-smooth, it still performs much better than PPO or A2C that are unable to learn the optimal policy function for such a small $dt$. We note that PINNs remains competitive with PPO and A2C when $O$ is relatively small (Pendulum and CartPole). However, as the domain size increases, PINNs begins to yield in performance compared to PPO. Increasing the number of samples in the dataset can improve its performance, but it means more computational resources are required.

| environment | method | mean | std |
|---|---|---|---|
| Pendulum | DP | 4133.71 | 433.02 |
| | A2C | 2180.22 | 766.25 |
| | PPO | 3273.51 | 906.41 |
| | PINNs | 3860.2 | 511.0 |
| CartPole | A2C | 1697.15 | 398.81 |
| | PPO | 5000.0 | 0.0 |
| | PINNs | 5000.0 | 0.0 |
| CartPole Swing-Up | A2C | 90.87 | 0.73 |
| | PPO | 970.63 | 130.3 |
| | PINNs | 723.3 | 175.16 |
| Acrobot | PPO | 1387.3 | 294.1 |
| | PINNs | 506.4 | 180.8 |

Table 1: Mean and standard deviation of the cumulative reward for different methods.

# 6 CONCLUSION

In this article, we consider the problem of finding the viscosity solutions of the deterministic HJB equation with neural network solvers. We propose a general scheme, which relies on solving a series of different PDE equations depending on $\epsilon$. When $\epsilon = 0$, the original HJB equation is retrieved. This framework gives flexibility on how $\epsilon$ are updated. In our experiments, we have shown that our scheme is able to learn the optimal value function with different $\epsilon$-schedulers for Pendulum. However, PINNs still struggle to match the performances of DTRL algorithms on larger domains. First reason that prevents it from scaling is its difficulty to match preciselsy the solution at the points of non-smoothness, which causes a bad control at those points. The second one is its restriction to uniform sampling across the domain, not allowing it to train more on more informative points unlike RL. In future works, we plan to get over those limitations. Integrating the adaptive sampling methods can help to improve sample efficiency and considering more sophisticated neural networks can help with better approximating non-smooth areas. Another limitation of our work is that it assumes that the dynamics are continuous ($f \in C(O)$), which is an important assumption for proving uniqueness of a viscosity solution. Thus, the approach considered in this paper cannot be applied to the case of non-continuous dynamics in the straightforward way. As the latter case is very important in real life applications, it should be studied in the future work. Finally, an interesting research perspective is to add model learning and actor/critic paradigm into our algorithm to explore the unknown dynamics case and enable the training based on trajectories.

Despite that, training a PINNs agent remains technically very challenging and it is not yet possible to cope with optimal control tasks that are as large as those dealt by DTRL. Yet, it is our goal to design agents suitable for solving large CTRL tasks and we provide some recipes to make it eventually as easy and accessible as training a DQN or PPO for DTRL.

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

# A   APPENDIX

## A.1   OPTIMAL CONTROL BACKGROUND

We use the same set-up as in Section 3.1. Further, we generalize the problem to the situations when exiting the domain $O$ is possible and we state some known theoretical results from the literature.

### A.1.1   GENERAL FORMALISM

In addition to the notions defined in Section 3.1, we also introduce some additional notations related to exiting the domain. Let $\tau$ denote the exit time. At time $\tau$, we have $x(\tau) \in \bar{O}$. Thus, let us define the exit reward $R : \partial O \to \mathbb{R}$, which is obtained at the boundary points when control is pushing the system out of $\bar{O}$. Under these notations, we redefine the reinforcement functional:

$$J(x_0; u(t)) = \int_0^\tau \gamma^t r(x(t), u(t)) dt + \gamma^\tau R(x(\tau)) \tag{14}$$

and the value function

$$V(x) = \sup_{u(t) \in U} J(x; u(t)). \tag{15}$$

Note that, when $R(x) \to -\infty$, then $\tau \to \infty$, which brings us back to the problem considered in Section 3.1

### A.1.2   HAMILTON-JACOBI-BELLMAN EQUATION

Similar to Section 3.2, the similar result holds for the value function defined with equation 15:

**Theorem A.1** *(Hamilton-Jacobi-Bellman). If the value function $V$ is differentiable at $x$, then the Hamilton-Jacobi-Bellman (HJB) equation holds at any $x \in O$:*

$$V(x) \ln(\gamma) + \sup_{u \in U} \left\{ \nabla_x V(x)^T f(x, u) + r(x, u) \right\} = 0. \tag{16}$$

*When $O \subset \mathbb{R}^d$, $V$ also statisfies the following boundary conditions:*

$$V(x) \geq R(x) \quad \text{for } x \in \partial O \tag{17}$$

### A.1.3   UNIQUENESS OF VISCOSITY SOLUTIONS

In this section, we state more formally the uniqueness result that holds for viscosity solutions and the additional assumptions under which it is verified. This section presents the short summary of the main theoretical results from Fleming & Soner (2006).

**Assumption A.1**   *(i)  $U$ is bounded,*

   *(ii)  $f, r$ are bounded, $f$ is continuous on $\mathbb{R}^d \times U$ and $r$ is uniform continuous on $\mathbb{R}^d \times U$,*

   *(iii)  there exists $L_f$, such that $\|f(x, u) - f(y, u)\| \leq L_f \|x - y\|$ for any $x, y \in \mathbb{R}^d$.*

**Theorem A.2**  *Given Assumption A.1, the value function $V$ is uniformly continuous and bounded in $\mathbb{R}$ and then it is a unique viscosity solution of the HJB equation equation 4.*

This theorem is given for the case when $O = \mathbb{R}^d$ and therefore there is no boundary condition. The other cases are discussed in the next sections.

The proof of Theorem A.2 consists of several parts. First, one can prove that under Assumption A.1, the value function is indeed uniform continuous and bounded. Then, one can show that it is a viscosity solution due to the dynamic programming principle that holds in continuous time case as well. The uniqueness comes from the comparison principle. It states that under Assumptions A.1, if $W$ and $V$ are viscosity subsolution and supersolution respectively and are bounded and uniformly continuous functions, then $W \leq V$. The comparison principle implies that if such $W$ and $V$ are viscosity solutions (both subsolutions and supersolutions), then $W \leq V$ and $W \geq V$, therefore $W \equiv V$. Thus, $V$ is a unique viscosity solution of the HJB equation in $\mathbb{R}^d$.

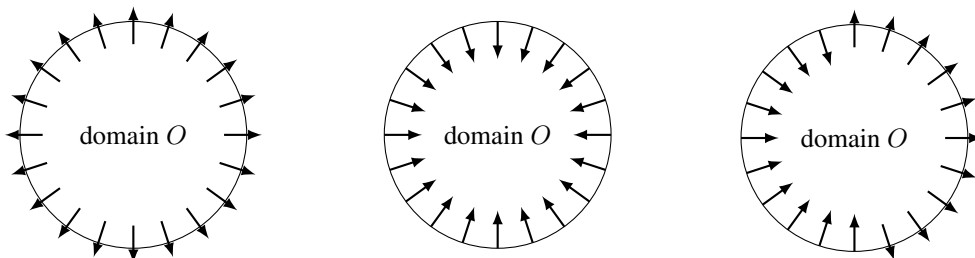

Figure 3: Boundary conditions. Case 1(left), Case 2 (middle) and Case 3(right).

### A.1.4 Viscosity for Different Boundary Conditions

When $O \not\equiv \mathbb{R}^d$, the additional assumptions on $O$ are required.

**Assumption A.2** *For any $x \in \partial O$ and its normal vector $\eta(x)$*

    *(i) $\exists u(x) \in U$ with $f(x, u(x))^T \eta(x) < 0$;*

    *(ii) $\exists u(x) \in U$ with $f(x, u(x))^T \eta(x) > 0$.*

**Assumption A.3 (Regularity condition)** *There exist $\epsilon_0, r > 0$ and $\hat{\eta}(x)$, a bounded and uniform continuous map of $\bar{O}$, satisfying*

$$B(x + \epsilon\hat{\eta}(x), r\epsilon) \subset O, \quad \forall x \in O, \epsilon \in (0, \epsilon_0] \tag{18}$$

*with $B(x, r) = \{y \in \mathbb{R}^d : \|x - y\| < r\}$.*

In this section, we cover three different cases of boundary conditions that appear in control problems when $O \subset \mathbb{R}^d$, which are illustrated in Figure 3.

*Case 1* If the system exits at any boundary point $x \in \partial O$ once the boundary is reached, *e.g.* it is the case when the exit reward $R(x)$ is sufficiently high to prefer to leave the area, *e.g.* when $R(x) \equiv R \geq r$ for any $x \in \partial O$ and $r \geq r(\bar{x}, u(\bar{x}))$ for any $\bar{x}, u(\bar{x})$. Then, the boundary condition is described with the following equation:

$$V(x) - R(x) = 0 \quad \forall x \in \partial O. \tag{19}$$

The uniqueness result holds due to the comparison principle that states that under Assumptions A.1 and provided that $W$ and $V$ are bounded and uniform continuous functions, if $W$ and $V$ are viscosity subsolution and supersolution respectively, then $\sup_{x \in \bar{O}}(W(x) - V(x)) \leq \sup_{x \in \partial O}(W(x) - V(x))$. The existence of such value function is assured with Assumption A.2-(ii).

*Case 2* If the system never exits the control domain. Let us denote the external normal vector at point $x \in \partial O$ as $\eta(x)$, then this boundary can be expressed as $f(x, u^*(x))^T \eta(x) \leq 0$ for any $x \in \partial O$. The optimal policy is not known a priori and thus it is hard to verify this constraint. In Fleming & Soner (2006); Soner (1986), it was shown that it can be reformulated as:

$$- H(x, W, \nabla_x W + \alpha\eta(x)) \leq 0 \quad \forall \alpha \leq 0, x \in \partial O. \tag{20}$$

This allows to extend Definition 3.1.

**Definition A.1 (Constrained viscosity solution)** *$W \in C(\bar{O})$ is called a constrained viscosity solution of the HJB equation equation 16 if it is a viscosity subsolution in $O$ and a viscosity supersolution in $\bar{O}$, i.e. if $\forall \psi \in C^1(\bar{O})$ and $\forall x \in \bar{O} \cup \arg\min\{(W - \psi)(x) : x \in \bar{O}\}$ with $W(x) = \psi(x)$, we have:*

$$H(x, \psi(x), \nabla_x \psi(x)) \geq 0.$$

It is also possible to prove that there exists a continuous value function provided that Assumption A.2(i) holds and the set of admissible actions is not empty for any state of the system. Under the additional Assumption A.3, there exists a unique constrained viscosity solution (see Soner (1986)).

*Case 3* If there exists a subset of points of the boundary at which the system exits the control domain. This is the same boudary condition considered in Munos (2000). This boundary is formulated as follows:

$$R(x) - V(x) \leq 0 \quad \forall x \in \partial O. \tag{21}$$

However, this boundary is not sufficient to have uniqueness, therefore we redefine viscosity for this inequality constraint equation 21.

**Definition A.2 (Viscosity solution with the boundary condition equation 21)** •

$W \in C(\bar{O})$ *is a viscosity subsolution of the HJB equation in $O$ with the boundary condition equation 21 if it is a viscosity subsolution in $O$ and $\forall \psi \in C^1(\bar{O})$ and $\forall x \in \partial O$ local maximum of $W - \psi$ such that $W(x) = \psi(x)$, we have:*

$$\min\{H(x, \psi(x), \nabla_x \psi(x)), R(x) - W(x)\} \leq 0$$

- $W \in C(\bar{O})$ *is a viscosity supersolution of the HJB equation in $O$ with the boundary condition equation 21 if it is a viscosity supersolution and $\forall \psi \in C^1(O)$ and $\forall x \in \partial O$ local minimum of $W - \psi$ such that $W(x) = \psi(x)$, we have:*

$$\max\{H(x, \psi(x), \nabla_x \psi(x)), R(x) - W(x)\} \geq 0$$

- *If $W \in C(\bar{O})$ is a viscosity subsolution and a supersolution with the boundary condition equation 21 then it is a viscosity solution with the boundary condition equation 21.*

It is easy to check that when equation 21 is verified then a viscosity subsolution $W(x)$ in $O$ is a viscosity subsolution with the boundary condition equation 21. However, when $W(x) > R(x)$ for some point $x \in \partial O$ then definition A.2 imposes an additional constraint that $W(x)$ should be a viscosity supersolution at such boundary points. Then similarly to Case 2, boundary condition equation 6 should be also satisfied, which can be interpreted as the system not being able to exit at those points. Similarly to Case 2, there is a uniqueness result:

**Theorem A.3** *Let us assume that Assumptions A.1-A.3 hold, then the value function $V$ is in $C(\bar{O})$ and it is the unique viscosity solution of the HJB equation in $O$ with the boundary condition equation 21.*

The proof of this theorem can be found in Fleming & Soner (2006); Cannarsa et al. (1991).

We choose to distinguish 3 different cases as it creates 3 different ways of approaching boundary conditions in practice. Indeed, equation 19, equation 20 and equation 21 produce different boundary losses for PINNs, *i.e.* $\mathcal{L}_{\partial O}$. However, note that Case 1 and Case 2 are subcases of Case 3.

Finally, some of the assumptions can be relaxed and it is possible to obtain more general uniqueness results (see Fleming & Soner (2006); Cannarsa et al. (1991); Ishii (1984)). However, the assumptions mentioned earlier are verified for the large class of control problems that appear in practice, like classical control or MuJoCo problems with no contacts (Todorov et al., 2012). Dealing with more general dynamics should be tackled in the future works.

## A.2 Intuition for Viscosity Solutions

In this section, we aim at providing the intuition behind the viscosity solutions. For that, we draw some parallels between DTRL and CTRL.[4]

Let us consider the DTRL formulation of the problem. We know that the optimal value function $V$ in DTRL should satisfy the Bellman equation

$$V(x) = \max_u \{r(x, u) + \gamma \sum_{x'} p(x'|x, u) V(x')\}. \tag{22}$$

From that, we can introduce the Bellman operator as

$$T(\psi)(x) = \max_u \{r(x, u) + \gamma \sum_{x'} p(x'|x, u) \psi(x')\} \tag{23}$$

---

[4]This section is based on `https://benjaminmoll.com/wp-content/uploads/2020/02/viscosity_for_dummies.pdf`.

where $\psi$ is an arbitrary function defined on the state space. This operator is known to be monotonic, *i.e.* for any functions $\psi, \psi'$ we have

$$\psi \geq \psi' \Rightarrow T(\psi) \geq T(\psi'). \tag{24}$$

Moreover, from Eq. equation 22 follows that $V$ should satisfy $V = T(V)$. Therefore, from Eq. equation 22-equation 24 we get the alternative definition for the solution of the Bellman equation 22.

**Definition A.3** *Let* $V \in C(O)$, *then* $V$ *is the optimal value function if and only if*

- $\forall \psi \in C^1(O)$ *such that* $\psi \geq V$

$$\max_u \{r(x,u) + \gamma \sum_{x'} p(x'|x,u)\psi(x')\} \geq V(x), \forall x \in O,$$

- $\forall \psi \in C^1(O)$ *such that* $\psi \leq V$

$$\max_u \{r(x,u) + \gamma \sum_{x'} p(x'|x,u)\psi(x')\} \leq V(x), \forall x \in O.$$

This definition can be seen as the discrete-time version of the viscosity solution definition. Therefore, in the discrete-time case, satisfying the fixed point equation is equivalent to satisfying the "discrete-time" viscosity solution definition.

As mentionned in the paper, $V \in C(O)$ can be non differentiable at some points of $O$, thus it is impossible to verify whether HJB equation is satisfied everywhere. Therefore, the main idea behind viscosity solutions is to replace $V$ by some smooth functions where $V$ is non differentiable.

In the following, first, we suppose that $V$ is differentiable everywhere and we show a connection between Hamilton-Jacobi-Bellman equation and Bellman equation. Then, for the case when $V$ is non smooth, we replace $V$ by a smooth function and we show that it is possible to derive the notions of viscosity super/subsolutions.

Let us discretize our continous-time problem with a time-step $dt$. For simplicity, we consider that for any $x \in O$ there exists an optimal control $u^* = \pi(x) \in U$ so that $V(x) = J(x; u^*)$. Therefore, we replace $\sup$ with $\max$ in the definition of the value function, though it is possible to show that the next results also hold in case of $\sup$. From the definition of the value function, we get

$$V(x(t)) = \max_u \left\{ \int_t^{t+dt} \gamma^{(s-t)} r(x(s), u(s)) ds + \gamma^{dt} V(x(t+dt)) \right\}$$

$$= \max_u \left\{ \int_t^{t+dt} \gamma^{(s-t)} r(x(s), u(s)) ds + e^{dt \ln(\gamma)} V(x(t+dt)) \right\}$$

$$\approx \max_u \left\{ r(x(t), u(t)) dt + e^{dt \ln(\gamma)} V(x(t+dt)) \right\}$$

$$\approx \max_u \left\{ r(x(t), u(t)) dt + (1 + \ln(\gamma) dt) V(x(t+dt)) \right\}$$

So we derive this discrete-time dynamic programming problem:

$$V(x_t) = \max_u \left\{ r(x_t, u) dt + (1 + \ln(\gamma) dt) V(x_{t+dt}) \right\}, \tag{25}$$

where $x_{t+dt} = f(x_t, u) dt + x_t$.

Let us suppose that $V$ is differentiable for all $x \in O$ and that $dt \in \left(0, -\frac{1}{\ln(\gamma)}\right)$. By subtracting $(1 + \ln(\gamma) dt) V(x_t)$ from both sides of Eq. equation 25 and then dividing by $dt$, we obtain

$$-\ln(\gamma) V(x_t) = \max_u \left\{ r(x_t, u) + \left(\frac{1}{dt} + \ln(\gamma)\right) (V(x_{t+dt}) - V(x_t)) \right\}.$$

If $dt$ goes toward 0, we have

$$\ln(\gamma) V(x_t) = -\max_u \left\{ r(x_t, u) + \nabla_x V(x_t)^T f(x_t, u) \right\}.$$

This is exactly the Hamilton-Jacobi-Bellman equation, the continuous time equivalent of the Bellman equation.

Now, let us assume that $V$ is non differentiable. As mentionned before, $V$ should be replaced by a smooth function at the points where $\nabla_x V$ does not exist. Let $\psi$ be an arbitrary smooth function on $O$ such that $V - \psi$ has a local maximum at $x_t$ and $V(x_t) = \psi(x_t)$. Therefore, $V \leq \psi$ in a neighborhood of $x_t$. If $1 + \ln(\gamma)dt > 0$, then

$$V(x_t) = \max_u \left\{ r(x_t, u)dt + (1 + \ln(\gamma)dt)V(x_{t+dt}) \right\}$$

$$\leq \max_u \left\{ r(x_t, u)dt + (1 + \ln(\gamma)dt)\psi(x_{t+dt}) \right\}$$

Let us subtract $(1 + \ln(\gamma)dt)\psi(x_t)$ from both sides and use $\psi(x_t) = V(x_t)$, as a result we have

$$- \ln(\gamma)V(x_t)dt \leq \max_u \left\{ r(x_t, u)dt + (1 + \ln(\gamma)dt)(\psi(x_{t+dt}) - \psi(x_t)) \right\}$$

Then, let us divide by $dt$ and let $dt$ goes toward 0, we have

$$- \ln(\gamma)V(x_t) \leq \max_u \left\{ r(x_t, u) + \nabla_x\psi(x_t)^T f(x_t, u) \right\}$$

$$\Leftrightarrow \quad \ln(\gamma)V(x_t) - \max_u \left\{ r(x_t, u) + \nabla_x\psi(x_t)^T f(x_t, u) \right\} \leq 0$$

$$\Leftrightarrow \quad H(x_t, \psi(x_t), \nabla_x\psi(x_t)) \leq 0.$$

This gives us the definition of a viscosity subsolution.

It is possible to obtain the definition of a viscosity supersolution in a like manner, by performing the same derivations for an arbitrary $\psi \in C^1(O)$ such that $V - \psi$ has a local minimum in $x_t$ and $V(x_t) = \psi(x_t)$. In both cases, we use the monotonicity of $\max_u \left\{ r(x_t, u)dt + (1 + \ln(\gamma)dt)\psi(x_{t+dt}) \right\}$ in the function $\psi$, which is a counterpart of the Bellman operator in Definition A.3.

Thus, we recover the defition of a viscosity solution. The intuition is whenever a solution $V$ of the HJB equation is non differentiable at some point $x \in O$, it should also satisfy other conditions imposed by viscosity for it to be a proper value function. In this way, the viscosity property serves as a regularizer to help to eliminate "bad" solutions of the HJB equation.

### A.3 DYNAMIC PROGRAMMING

Further, we consider only FEM based dynamic programming proposed in Munos (2000). In the FEM case, we use a triangulation $\Sigma^\delta$ to cover the state space. It is also possible to discretize the control space, denoted by $U^\delta$. The vertices of the triangulation $\Sigma^\delta$ are denoted $\{\xi_1, \xi_2, ..., \xi_{N_\delta}\}$ with $N_\delta \in \mathbb{N}$. In this setting, $V$ is approximated by a piecewise linear function $V^\delta$. Thus, for $x \in Simplex(\xi_0, .., \xi_d)$, we have

$$V^\delta(x) = \sum_{i=0}^{d} \lambda_{\xi_i}(x)V^\delta(\xi_i)$$

where $\lambda_{\xi_i}(x)$ is the barycentric coordinates inside the simplex $(\xi_0, ..., \xi_d)$.

By using a FEM approximation scheme, the HJB equation is transformed into:

$$V^\delta(\xi) = \sup_{u \in U^\delta} [\gamma^{\tau(\xi, u)} V^\delta(\eta(\xi, u)) + \tau(\xi, u)r(\xi, u)]$$

where $\eta(\xi, u) = \xi + \tau(\xi, u)f(\xi, u)$ and $\tau(\xi, u)$ is a time discretization function that should satisfy:

$$\exists k_1, k_2 > 0, \forall \xi \in \Sigma^\delta, \forall u \in U^\delta, k_1\delta \leq \tau(\xi, u) \leq k_2\delta$$

If $F^\delta$ is defined as $F^\delta[\phi](\xi) = \sup_{u \in U^\delta} [\gamma^{\tau(\xi, u)} \sum_{i=0}^{d} \lambda_{\xi_i}(\eta(\xi, u))\phi(\xi_i) + \tau(\xi, u)r(\xi, u)]$, it is possible to show that $F^\delta$ satisfies a contraction property, and since $V^\delta(\xi) = F^\delta[V^\delta](\xi)$ holds, dynamic programming techniques can be applied to compute $V^\delta$. Moreover, it can be proved that $V^\delta \underset{\delta \to 0}{\longrightarrow} V$ uniformly on any compact of the state space. With this method, one can derive algorithms that converge towards $V$, without even knowing the dynamics of the system. Thus, this is one of the approaches that allows us to find a viscosity solution of the HJB equation (see Munos (2000) for more details).

## A.4 INTRODUCTION TO PINNs

Here, we provide a short introduction to PINNs for those readers who are not familiar with this method. The adaptation of PINNs to solving the HJB equation in the viscosity sense is covered in Section 4.2.

To solve a differential equation

$$F(x, W(x), \nabla_x W(x), \nabla^2_{xx} W(x)) = 0, \quad W : \bar{O} \to \mathbb{R}, x \in O, \tag{26}$$

with $K_1$ equality boundary conditions

$$B_i(x, W(x), \nabla_x W(x), \nabla^2_{xx} W(x)) = 0, \quad x \in \partial O, i \le K_1, \tag{27}$$

and $K_2$ inequality boundary conditions

$$G_i(x, W(x), \nabla_x W(x), \nabla^2_{xx} W(x)) \le 0, \quad x \in \partial O, i \le K_2. \tag{28}$$

one can assume that $W(x)$ lies in the class of functions $\mathcal{F}_\theta = \{f_\theta(x) = NN(x, \theta) : \theta \in \Theta\}$ represented by neural networks of a fixed architecture and parametrized with weights $\theta \in \Theta$. If it is the case then there exists $\theta$ such that $W(x, \theta)$ should satisfy equation 26 and thus minimize the loss

$$\mathcal{L}_{PDE}(\theta) = \frac{1}{N_F} \sum_{i=1}^{N_F} \left(F(x_i, W(x_i, \theta), \nabla_x W(x_i, \theta), \nabla^2_{xx} W(x_i, \theta))\right)^2 \quad \forall x_i \in \mathcal{S}_u(O, N_F), \tag{29}$$

with $\mathcal{S}_u(O, N_F)$ denoting a sample of $N_F$ points drawn uniformly from $O$. If the solution $W(x, \theta)$ should satisfy some additional boundary constraints then it should also minimize the boundary losses for all $k \le K_1$ and $k' \le K_2$

$$\mathcal{L}_{B_k}(\theta) = \frac{1}{N_B^k} \sum_{i=1}^{N_B^k} \left(B_k(x_i, W(x_i, \theta), \nabla_x W(x_i, \theta), \nabla^2_{xx} W(x_i, \theta))\right)^2 \quad \forall x_i \in \mathcal{S}_u(\partial O, N_B^k) \tag{30}$$

$$\mathcal{L}_{G_{k'}}(\theta) = \frac{1}{N_G^{k'}} \sum_{i=1}^{N_G^{k'}} \left(\left[G_{k'}(x_i, W(x_i, \theta), \nabla_x W(x_i, \theta), \nabla^2_{xx} W(x_i, \theta))\right]^+\right)^2 \quad \forall x_i \in \mathcal{S}_u(\partial O, N_G^{k'}), \tag{31}$$

where $[f(x)]^+ = \max\{f(x), 0\}$.

To put everything together, when solving a PDE in a PINNs-like manner, one should train a neural network $W(x, \theta)$ that minimizes:

$$\mathcal{L}(\theta) = \mathcal{L}_{PDE}(\theta) + \sum_{k=1}^{K_1} \lambda_k \mathcal{L}_{B_k}(\theta) + \sum_{k=1}^{K_2} \lambda'_k \mathcal{L}_{G_k}(\theta). \tag{32}$$

where $\lambda_k, \lambda'_k > 0$ are some mixing coefficients for different boundary conditions.

## A.5 EXPERIMENTAL RESULTS. SUPPLEMENTARY

### A.5.1 DYNAMIC PROGRAMMING EXPERIMENTAL RESULTS

In this section, we present the results obtained with one of the algorithms proposed in Munos (2000). First, a grid is built by dividing each axis by $N$ points. Then, we use the Delaunay's triangulation over the grid and apply the Value Iteration algorithm (VI) to the FEM-MDP derived in Munos (2000).

We set $\delta = \frac{1}{N}$, $\delta$ being the discretization step. The stopping criterion used at the step $n$ is $\|V_n - V_{n-1}\|_\infty \le \epsilon$ where $\epsilon$ is a chosen tolerance. In our experiments we work with $\epsilon = 10^{-5}$.

When $\delta$ goes towards 0, our approximated value function, $V^\delta$, converges towards the true value function. In our case, $\delta \to 0$ is equivalent to $N \to +\infty$. Empirically, we can see in Figure 4 that this property is satisfied. Indeed, as we increase $N$, we obtain a more accurate $V^\delta$, and as a result, a better control that leads to a higher cumulative reward.

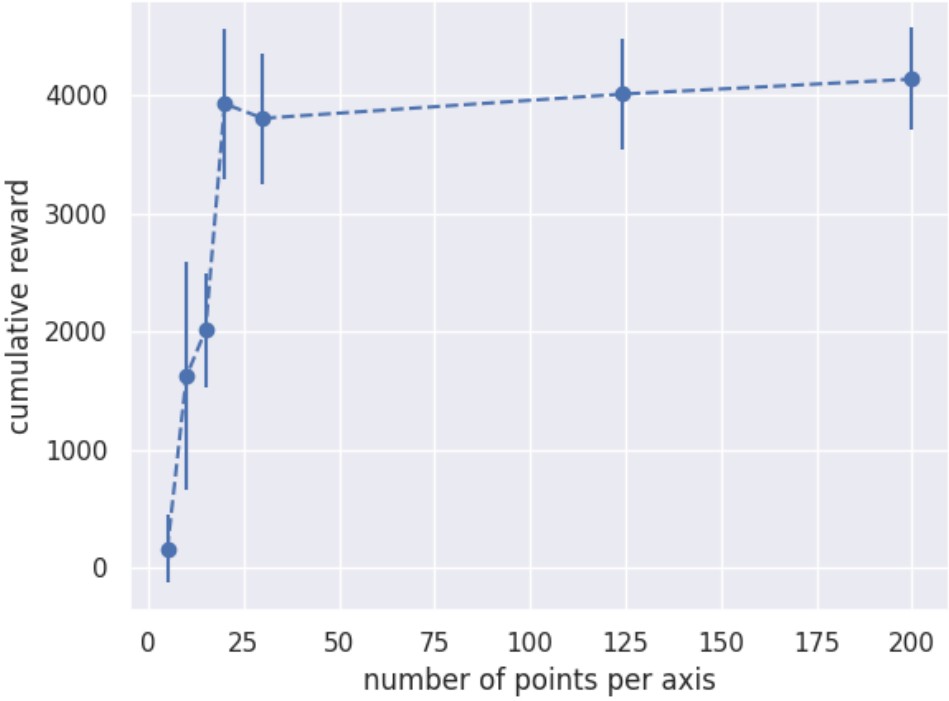

Figure 4: The cumulative reward obtained on the inverted pendulum for different grid sizes $N$.

### A.5.2 COMPARISON OF $\epsilon$ SCHEDULERS

As mentioned in Section 4.2, we tested three kind of $\epsilon$-schedulers. All scheduler experiments have been performed with the parameters described in Section A.5.4, except mentioned so.

In Figure 5, we can see that the non-adaptive scheduler never leads to the true solution for $k_\epsilon = 0.9$, even with 75 epochs between each update.

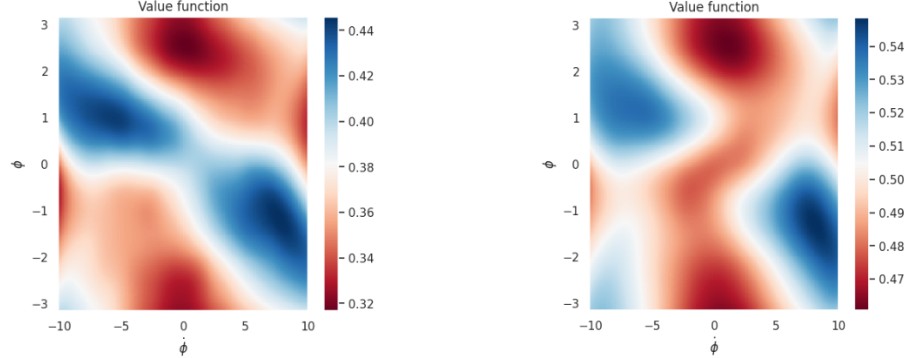

Figure 5: Value function obtained for non-adaptive scheduler. The parameters used are $k_\epsilon = 0.9, N_u = 25$ (left) and $k_\epsilon = 0.9, N_u = 75$ (right).

As a result, we try another scheduling strategy. This time, the loss is taken into account. It gives us a metric to know whether the convergence is achieved or not. Thanks to this scheduler, it is possible to approximate the true solution by starting from a small $\epsilon_0$ (e.g. $\epsilon_0 = 10^{-3}$). Nevertheless, it is preferable to start from a higher $\epsilon_0$ (e.g. $\epsilon_0 = 1$) to improve the algorithm stability. Indeed,

more $\epsilon$ is high, more $W^\epsilon$ is easy to approximate. This property can be observed empirically in Appendix A.5.3. Therefore, by starting from a high $\epsilon_0$, the risk of divergence is curbed, but the training time is increased since we need more iterations to reach the $\epsilon_n$ that are very near to 0. In Figure 6, we can see that if we start from $\epsilon_0 = 10^{-3}$ with $k_\epsilon = 0.9$ then it is possible to reach a good value function approximation. At the same time, the algorithm may diverge for some seeds (different random initializations). By trying to increase the convergence speed and setting $k_\epsilon = 0.1$ leads to a lot of instability during the training phase. In Figure 7 we can note that for both $\epsilon_0 = 1, \epsilon_0 = 10^{-3}$, the algorithm may diverge.

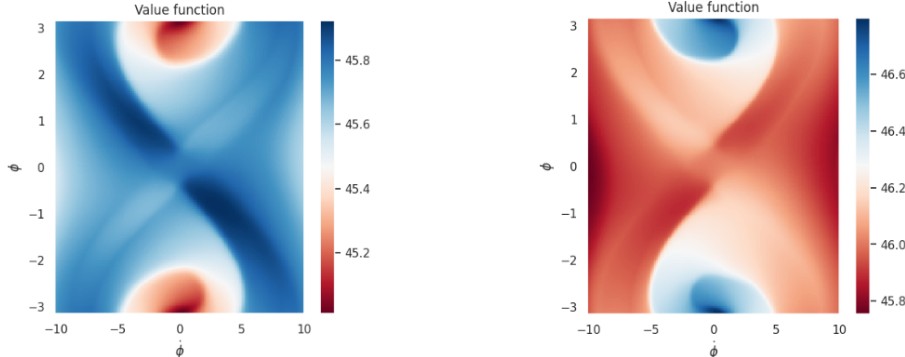

Figure 6: Value function for $k_\epsilon = 0.9$ with $\epsilon_0 = 1$ (left) and $\epsilon_0 = 10^{-3}$ (right).

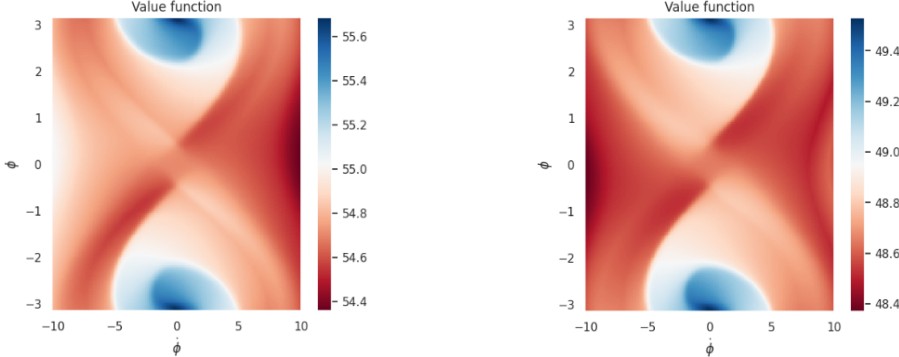

Figure 7: Value function for $k_\epsilon = 0.1$ with $\epsilon_0 = 1$ (left) and $\epsilon_0 = 10^{-3}$ (right).

Finally, we have mixed the two algorithms to keep the strength of both schedulers while their weaknesses are eliminated. Starting with the non-adaptive scheduler during the "easy" phase (the first $\epsilon_n$ for which $W^{\epsilon_n}$ are easy to approximate) allows us to rapidly reach a small $\epsilon_n$ and then the adaptive scheduler is used to reduce the risk of divergence. We have tested the hybrid algorithm on 8 different seeds and our model has reached convergence on every attempt. Our final choice of parameters is $\epsilon_0 = 1, k_\epsilon = 0.1, N_u = 10$ for 5 updates of $\epsilon$ then we switch to the adaptive scheduler and we set $k_\epsilon = 0.9, n_\epsilon = 7$.

### A.5.3 CONVERGENCE OF PINNS FOR A FIXED $\epsilon$

In this section, we provide the results that we have obtained for different fixed $\epsilon$. We have used the same parameters as in table 2. On figure 8, it is clear that it is easier for PINNs to approach $W^\epsilon$ when $\epsilon$ is high enough. Therefore, we assume that starting our $\epsilon$ scheduler from $\epsilon_0 = 1$ leads to a more stable convergence. That is one of the reasons we designed the hybrid scheduler: to improve the stability while maintaining a good speed.

### A.5.4 BEST HYPERPARAMETERS

The best performing hyperparameters are gathered in Table 2.

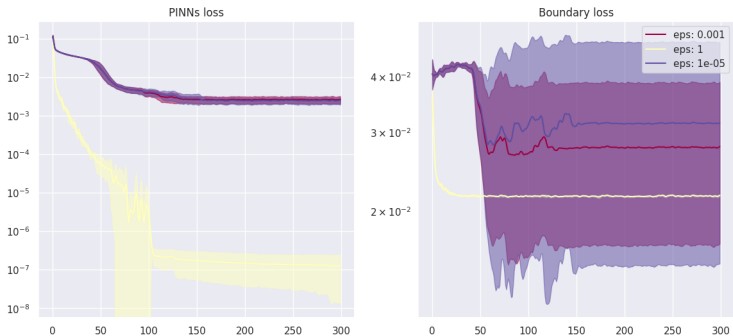

Figure 8: $W^\epsilon$ pinns loss and boundary loss for $\epsilon = 1, 10^{-3}, 10^{-5}$

| Environment | Names | Hyperparameters | values |
|---|---|---|---|
| Shared | batch size | $N_S$ | 100 |
| | learning rate | $\nu$ | 0.00085 |
| | patience adaptive scheduler | $n_\epsilon$ | 1 |
| | boundary loss coefficient | $\lambda$ | $10^{-1}$ |
| | starting $\epsilon$ | $\epsilon_0$ | 1 |
| | number of epochs between $\epsilon$ updates | $N_u$ | 10 |
| | non-adaptive scheduler coefficient | $k_\epsilon$ | 0.1 |
| | adaptive scheduler coefficient | $k_\epsilon'$ | 0.99 |
| | number of $\epsilon$ updates with non-adaptive scheduler | $N_\epsilon$ | 5 |
| Pendulum | number of sampled points | $N_D$ | 20000 |
| | reg loss coefficient | $\lambda_R$ | $10^{-3}$ |
| Cartpole | number of sampled points | $N_D$ | 30000 |
| | reg loss coefficient | $\lambda_R$ | $10^{-3}$ |
| Acrobot | number of sampled points | $N_D$ | 30000 |
| | reg loss coefficient | $\lambda_R$ | $10^{-2}$ |

Table 2: Hyperparameters for Algorithm 1.

