# OpenReview forum: "Learning HJB Viscosity Solutions with PINNs for Continuous-Time Reinforcement Learning"
_ICLR.cc/2024/Conference — Submitted to ICLR 2024_

### Official Review · Reviewer_DLUu · 2023-10-30

**Soundness:** 3 good
**Presentation:** 2 fair
**Contribution:** 3 good
**Rating:** 5
**Confidence:** 4

**Summary:**

This paper proposes a training scheme for neural networks to approximate
viscosity solutions of HJB equations. This presents a principled
approach to optimal control in continuous-time deterministic RL
settings, overcoming the curse of dimensionality incurred by existing
viscosity solution methods that rely on space
discretization. Experiments on classical control environments with
small timesteps are conducted, and the authors both demonstrate
desirable performance of their proposed method and identify the
remaining challenges.

**Strengths:**

I found the approach presented in the paper to be interesting. While
it is based on an existing approximation scheme for viscosity
solutions, I have never seen this applied to RL or general function
approximation methods for solving HJB equations. The approach is
sensible, and highlights an alternative paradigm to value learning
that replaces bootstrapping with PDE solving (using something like a
collocation method, as I understand it). I really appreciate how the
authors identified the remaining challenges surrounding their
approach, which seems to provide nice motivation and direction for
future work. I generally enjoyed learning about the proposed method,
and found the scheduling and regularization technique interesting.

**Weaknesses:**

The most major issues for me are:
1. The PINN method does not perform as well as existing DTRL methods
   (or at least PPO) in settings that require function
   approximation. While I appreciate that the authors give some
   insight about why that may be the case (and ideas for future
   research towards bridging the gap), I wish there was more
   motivating results: even if the highlighted issues with the
   approach are solved, what benefits should we expect to see relative
   to DTRL algorithms? Particularly, it would have been nice to see
   some results from the PINN method trained on much more data (i.e.,
   to overcome the uniform sampling issue), to visualize the benefits
   we can expect if we eventually find a more efficient training
   scheme.
2. While the $\epsilon$ schedules that are presented seem intuitively
   reasonable, it would have been nice to see more discussion and/or
   analysis about those in the paper. I understand that space is
   limited, but 5 pages are spent before the method is even
   presented. Is there a theoretically principled way to choose the
   $\epsilon$ schedule, or is there a principled way to quantitatively
   compare them?

When referring to the HJ-DQN algorithm (Kim et. al, 2021), it says
"this approach is limited to Lipschitz continuous control". I am
familiar with this paper so I know what this means, but I definitely
think it is worth clarifying further, since "Lipschitz continuous
control" can be interpreted in at least a few different ways. On this
note, is this really a limitation? Controls that vary smoothly in time
are often desirable.

Citations for Definition 3.1 and Lemma 3.1 should be given.

It should be made more clear why you call some HJB solutions "bad" vs
"good".

## Minor issues
Page 3, center, "An other" should be "Another".

Page 3, just above section 3, "estimations of value function" should
be "estimations of value functions".

Just above section 3.3, it says "the value function $V$ is often
non-smooth, and only continuous on $O$". I think it would read better
to be more explicit about the problem here, for instance,
by explicitly saying that the value function may be non-differentiable
on $O$, and therefore cannot solve the HJB equation in the familiar sense.

What is the purpose of Figure 2 (middle)?

Table 1 is hard to read, there should at least be horizontal lines
to separate environments.

**Questions:**

The paper says that the work of Darbon et. al 2023 "work with min-plus
algebra", what does that mean? What types of optimal control problems
is it not suitable for?

In equation 7, what is $\nabla^2_x$? Is that the Laplacian? Likewise,
just below, what is $\nabla^2_{xx}$? What is the significance of the
RHS in equation 7, can we choose other "perturbations" instead? How
can you know if the RHS in equation 7 converges to 0 when
$\epsilon\to 0$?

At the bottom of page 5, it says "dynamic programming approaches are
able to find the solutions of the HJB equation that are intrinsically
viscosity solutions" -- what does "intrinsically viscosity solutions" mean?

Why does PPO seemingly scale better than the PINNs method? Is this
just because PPO is specializing to regions of the state space visited
by nearly optimal policies, whereas the PINNs method has to train on
the whole space? How do the methods based on the Pontryagin maximum
principle that were mentioned in the related work compare?

---

> ### Author Response · Authors · 2023-11-20
> **Answer Part I**
>
> We thank the reviewer for carefully reading our paper and pointing out some typos. We are happy to learn that you appreciated the idea of the paper and our approach to the problem. Below we try to address some of the questions raised.
>
> > The PINN method does not perform as well as existing DTRL methods (or at least PPO) in settings that require function approximation. While I appreciate that the authors give some insight about why that may be the case (and ideas for future research towards bridging the gap), I wish there was more motivating results: even if the highlighted issues with the approach are solved, what benefits should we expect to see relative to DTRL algorithms? Particularly, it would have been nice to see some results from the PINN method trained on much more data (i.e., to overcome the uniform sampling issue), to visualize the benefits we can expect if we eventually find a more efficient training scheme.
>
> We understand your point, but the problem is not in quantity of the data. Indeed, we tried to train our algorithm on larger datasets but after **beyond**  a certain dataset size there is little to no improvement on the result. The true limiting factor is the usage of uniform sampling in the domain. Indeed, "the hard areas", e.g. zones where the value function is particularly non-smooth or where distinguishing the best action is hard, require more samples to be learned properly than other zones. Similarly, DTRL algorithms learn more efficiently if the replay buffer has a considerable number of samples corresponding to high return trajectories.
>
> Adaptive sampling approaches [RAR, R3] are emerging as a way to bypass this issue. However, those methods cannot be directly applied to our method. Such adaptive sampling methods may switch "focus" from one sub-area of $O$ to another throughout the entire training, which violates the uniform convergence of $V^{\epsilon} \to V$ and thus "breaks viscosity". It is not yet trivial how to do adaptive sampling while preserving viscosity, but it is what we want to explore in future work.
>
>
> > While the schedules that are presented seem intuitively reasonable, it would have been nice to see more discussion and/or analysis about those in the paper. I understand that space is limited, but 5 pages are spent before the method is even presented. Is there a theoretically principled way to choose the schedule, or is there a principled way to quantitatively compare them?
>
> We have provided in the paper a discussion related to schedulers and some complementary plots in Appendix A.5.2. In addition to that, as there is no viscosity metric that we could use to compare different schedulers, we cannot really quantitatively compare them on how well they find the viscosity solutions. The useful metric that we actually can apply is the value of right hand side of Eq. 7, i.e. $\text{Tr}(\nabla^2 W^{\epsilon}(x, \theta))$ or $\delta (\epsilon, \theta)$. However, for every scheduler, it is possible to choose the hyper-parameters that can lead to the convergence of $\text{Tr}(\nabla^2 W^{\epsilon}(x, \theta)) \to 0$ if $\epsilon \to 0$. Finally, for every scheduler we were able to find the set of hyperparameters that manages to achieve the good convergence, but as we observed if those hyperparameters are not chosen carefully then some schedules fail more than the others. According to our experiments, hybrid is the most robust to the choice of hyperparameters.
>
> > When referring to the HJ-DQN algorithm (Kim et. al, 2021), it says “this approach is limited to Lipschitz continuous control”. I am familiar with this paper so I know what this means, but I definitely think it is worth clarifying further, since “Lipschitz continuous control” can be interpreted in at least a few different ways. On this note, is this really a limitation? Controls that vary smoothly in time are often desirable.
>
> HJ-DQN is a good paper, but it is not exactly what we are looking at. Control that vary smoothly in time are desirable in critical applications such as robotic, self-driving cars. However, by constraining too much the set of admissible controls we may fail to find the optimal one. Moreover, there are systems for which the control varying smoothly in time is not feasible. Even though, we consider the control from the discrete space for now, we try to approach the problem of finding optimal control in a general way as possible and therefore without imposing too many constraints on the control space. Going to continuous control is important and interesting future work. At the same time, as we chose to work with more general settings in which $Q$ function degenerates, we cannot leverage on the works that are based on Q-learning and DQN-based algorithms, while HJ-DQN can be a way to adapt a lot of previous works on discrete-time to the continuous-time setting.
>
> We will change the text in our paper to better describe and compare to HJ-DQN.

---

> > ### Author Response · Authors · 2023-11-20
> > **Answer Part II**
> >
> > > Citations for Definition 3.1 and Lemma 3.1 should be given.
> >
> > Thank you for pointing it out! In the paper we have provided the citations for each corresponding notion, but after their introduction, we will move the citations before introducing those notions to make the sources more explicit.
> >
> > > It should be made more clear why you call some HJB solutions "bad" vs "good".
> >
> > We call viscosity solutions as "good" ones as they are the one that correspond to the optimal value function, while all other solutions are "bad". We will make it more explicit in the paper.
> >
> > > The paper says that the work of Darbon et. al 2023 "work with min-plus algebra", what does that mean? What types of optimal control problems is it not suitable for?
> >
> > This paper assumes the linear dynamics and quadratic rewards settings, similar to LQR. Restricting their problems to this settings helps them to propose NN architectures with viscosity guarantees. As we have already mentioned, we would like to avoid introducing strong assumptions on the shape of dynamics function, we only assume the smoothness, which is important for the uniqueness of the viscosity solution. Let us also note that the dynamics of the real physical systems are often nonlinear and it is the case of pendulum task as well: $\frac{d^2 \theta}{dt^2} = - \frac{g}{l} \sin \theta$. We will make it more clear in the related works section of the paper.
> >
> > >In equation 7, what is $\nabla^2_x$? Is that the Laplacian? Likewise, just below, what is $\nabla^2_{xx}$? What is the significance of the RHS in equation 7, can we choose other "perturbations" instead? How can you know if the RHS in equation 7 converges to 0 when $\epsilon \to 0$?
> >
> > Both $\nabla^2_x$ and $\nabla^2_{xx}$ denote the same thing ($\nabla^2_x$ being the typo, in most places we use $\nabla^2_{xx}$), which is the Hessian, thus $\text{Tr} \nabla^2_{xx} V(x)$ is a laplacian. We were considering before to use $\Delta V(x)$ as a laplacian, but this notation may be unfamiliar for some readers. We will make sure that there is no typo and less confusion about this notation in the paper. Thank you for noticing and pointing it out!
> >
> > >How can you know if the RHS in equation 7 converges to 0 when $\epsilon \to 0$?
> >
> > In case of non-adaptive scheduler we cannot know it indeed, unless we choose very small $k_{\epsilon}$, but then it can create too huge gap between $V^{\epsilon_n}$ and $V^{\epsilon_{n-1}}$. The adaptive and hybrid schedulers update $\epsilon$ in accordance with $\delta(\epsilon, \theta)$ to make sure that it decreases. As $\delta(\epsilon, \theta)$ can be seen as mean squared value of RHS in equation 7, then it also decreases with each iteration.
> >
> > > At the bottom of page 5, it says "dynamic programming approaches are able to find the solutions of the HJB equation that are intrinsically viscosity solutions" -- what does "intrinsically viscosity solutions" mean?
> >
> > We meant that some schemes are proven to converge to viscosity solutions for small enough discretization time step, even though the design of those schemes does not take viscosity into account. We will reformulate this sentence in the paper.
> >
> >
> > >Why does PPO seemingly scale better than the PINNs method? Is this just because PPO is specializing to regions of the state space visited by nearly optimal policies, whereas the PINNs method has to train on the whole space?
> >
> > Yes, we believe that this is PPO's advantage over our approach.
> >
> > >How do the methods based on the Pontryagin maximum principle that were mentioned in the related work compare?
> >
> > We haven't compare to those methods. Paper based on the Pontryagin maximum principle supposes that the cost function is convex (so that the reward function is concave). Pontryagin maximum principle gives a necessary condition as boundary value problem (BVP) that the value function should verify. In general, BVP admits several solutions which can be sub-optimal. It is mentioned that optimality can be guaranteed under some convexity assumptions but in practice it is difficult to verify such conditions globally. In this work, they supposed the optimality of the solutions. Their sampling method is different to ours. Indeed, they use numerical methods to compute trajectory on short time interval by solving the BVP and then use them as a dataset to train a NN. As a result, the training procedure is comparable to supervised learning, while ours is comparable to unsupervised learning. A limitation of this method is that, since the optimality condition is a BVP, it is necessary to define the boundary conditions as equality (Dirichlet and Neumann boundary conditions). As mentioned in our work, it's not always easy to define such boundary conditions.

---

> > > ### Comment · Reviewer_DLUu · 2023-11-22
> > >
> > > Thanks to the authors for the detailed response, which clarified some points of confusion for me. My understanding now is that the proposed PINN method is fundamentally limited by the requirement to update based on uniform samples in the state space (so even in the limit of infinite data, it cannot specialize enough to the most relevant part of the state space). This is a little underwhelming, since the main purpose of the approach is to overcome the curse of dimensionality, and the method cannot compete once the dimensionality is even moderate. The conclusion proposes future work towards resolving this issue with "adaptive sampling methods" -- I am not familiar with this line of work, but I'd be curious to see if such methods exist in the literature (outside of RL, I suppose). Is there any evidence that this is a feasible/promising direction?
> > >
> > > Re: HJ-DQN, I did not mean to compare your method with theirs, but I agree with what you said, and I appreciate the response.
> > >
> > > Re: "bad" HJB solutions -- isn't there a unique viscosity solution to the HJB equations you're considering (and this coincides with the value function)? If so, why do "bad" viscosity solutions exist?

---

> > > > ### Author Response · Authors · 2023-11-22
> > > >
> > > > >Thanks to the authors for the detailed response, which clarified some points of confusion for me. My understanding now is that the proposed PINN method is fundamentally limited by the requirement to update based on uniform samples in the state space (so even in the limit of infinite data, it cannot specialize enough to the most relevant part of the state space). This is a little underwhelming, since the main purpose of the approach is to overcome the curse of dimensionality, and the method cannot compete once the dimensionality is even moderate. The conclusion proposes future work towards resolving this issue with "adaptive sampling methods" -- I am not familiar with this line of work, but I'd be curious to see if such methods exist in the literature (outside of RL, I suppose). Is there any evidence that this is a feasible/promising direction?
> > > >
> > > > There is a number of recent papers trying to address the problem of PINNs training, in particular to learn high dimension PDEs. We list them bellow.
> > > >
> > > > The two following papers rely on tangent kernels to 1) identify the problem of PINNs convergence and 2) propose some adaptive sampling strategy based on the tangent kernel decomposition:
> > > > 1. WHEN AND WHY PINNS FAIL TO TRAIN: A NEURAL TANGENT KERNEL PERSPECTIVE. http://arxiv.org/abs/2007.14527
> > > > 2. PINNACLE: PINN ADAPTIVE COLLOCATION AND EXPERIMENTAL POINTS SELECTION. https://openreview.net/pdf?id=GzNaCp6Vcg
> > > >
> > > > These two other papers propose adaptive sampling strategies for PINNs with quite impressive results on their benchmarks. They basically look at the loss values to make a decision on sample selection for the next steps:
> > > >
> > > > 3. Mitigating Propagation Failures in Physics-informed Neural Networks using Retain-Resample-Release (R3) Sampling. http://arxiv.org/abs/2207.02338
> > > > 4. DeepXDE: A deep learning library for solving differential equations. https://arxiv.org/abs/1907.04502
> > > >
> > > > The following papers are particularly relevant as they address solving HJB with PINNS at high dimensions (but they both only consider HJB equations with linear-quadratic-Gaussian (LQG) control. These have a unique analytical solution, and so do not have the problem of converging toward the viscosity solution while avoiding all other possible non viscosity solutions as we do). The first paper shows that a $L_\infty$ norm is required for the loss to get PINNs to converge to a stable PDE solution. They propose an adversarial pattern to move an initial set of samples to high loss regions and then uses these samples for the training batch. Though not presented as an adaptive sampling strategy, it eventually boils down to one. They test their approach with HJB at dimension 250. The second very recent paper (previous week) compares to this work and proposes another approach where they perform a two level sampling, first by selecting randomly dimensions and them colocation points in these dimensions. They manage to train PINNS for HJB going as high as 100,000 dimensions.
> > > >
> > > > 5. Is $L^2$ Physics Informed Loss Always Suitable for Training Physics Informed Neural Network? https://proceedings.neurips.cc/paper_files/paper/2022/hash/374050dc3f211267bd6bf0ea24eae184-Abstract-Conference.html
> > > > 6. Tackling the Curse of Dimensionality with Physics-Informed Neural Networks - arXiv:2307.12306
> > > >
> > > > So globally all this research activity indicates that some form of adaptive sampling is a key to pushing PINNs training capabilities to higher dimensions and more complex problems. We are currently working on adopting some of the adaptive sampling methods and have actually started some experiments. However, the current results suggest that current adaptive sampling methods made for classical PINNs can be detrimental to the uniform convergence of solutions and thus breaking the viscosity. That is why, those results are not mature enough to be included in this paper and require further research on how to incorporate viscosity into the adaptive sampling.
> > > >
> > > > > Re: "bad" HJB solutions -- isn't there a unique viscosity solution to the HJB equations you're considering (and this coincides with the value function)? If so, why do "bad" viscosity solutions exist?
> > > >
> > > > To further clarify, as a viscosity solution is unique and corresponds to the value function, we call it a "good" solution (actually, only one solution corresponds to the value function, thus there is only one "good" solution), all other solutions, which are not viscosity solutions, are "bad" as they do not correspond to the value function. To avoid confusion, we will remove "good"/"bad" notions and stick to more explicit notions.

---

### Official Review · Reviewer_N25p · 2023-10-30

**Soundness:** 2 fair
**Presentation:** 3 good
**Contribution:** 2 fair
**Rating:** 3
**Confidence:** 4

**Summary:**

The authors introduced a numerical method for solving continuous time HJB equations in a deterministic control and reward setting. The method depends on convergence theory of viscosity solutions to value function, a core Lemma that the authors elaborated in a very clear and accessible way. With three different types of decaying schemes for a core parameter $\epsilon$, the authors used PINN/PDE solvers to solve a sequence of PDEs that approximate the interested optimal control problems. Numerical experiments demonstrate the introduced method is effective on some of the classic control tasks, although there are challenges to be addressed in the continuous time setting.

**Strengths:**

The authors give a very detailed and clear introduction of optimal control problems in continuous time and the necessity of involving viscosity solutions. This (and the Appendices) makes the manuscript very easy to understand for non-experts in control theory. The authors also provide plentiful details on the numerical experiments, such as how to blend different $\epsilon$ decaying schemes, what types of NN are more suitable for learning viscosity solution, etc. Numerical experiments, although for simple classic tasks, demonstrate acceptable improvements over other SOTA RL/optimal control methods.

**Weaknesses:**

The main weakness of this manuscript is its novelty and magnitude of its contributions in terms of both theoretical and experimental aspects. The details are listed below:

1. The main theoretical foundation is Lemma 3.1, which is a well-known results in optimal control community. For clarity, the authors have to spend more than half of the main context introduce this Lemma. This left the only novelty in the algorithm design where the authors introduced three $\epsilon$ decay schemes and the PDE solver (the latter of which has also been widely studied by PDE people). The overall significance of this work is therefore limited.

2. If the authors want to enhance the theoretical contribution of this work, I would prefer to see how they would address the convergence in more details. Lemma 3.1 is a general result, but in the PINN setting, what conditions on the NN can guarantee convergence and how to characterize approximation error or convergence speed, etc. These are challenging theoretical questions if the authors want to go multiple steps further than just citing Lemma 3.1 as the foundation of their method.

3. If the authors want to focus on the experimental aspect of their approach, I believe high dimensional control tasks may be the right playground for PINNs. The purpose of introducing NN in PDE community is for solving large dimensional problems. The value of PINN for optimal control problems are likely to be similar. Otherwise, with known dynamics and easily observable rewards (as in this work), it is relatively easy to come up with efficient functional forms to approximate value functions without PINNs.

**Questions:**

My suggestions for a more in-depth analysis from either theoretical or experimental perspective are in the above section.

---

> ### Author Response · Authors · 2023-11-20
> **Answer Part I**
>
> We thank the reviewer for carefully reading the paper!
>
> > The main theoretical foundation is Lemma 3.1, which is a well-known results in optimal control community. For clarity, the authors have to spend more than half of the main context introduce this Lemma.
>
> Indeed, this is a well-known result from optimal control and the paper mention that the proof can be found in Fleming & Soner (2006). We will make it more explicit for the final version.  As this result together with uniqueness of viscosity solutions is necessary for introducing viscosity into PINNs we consider it is very important to provide enough background on optimal control.
>
> > This left the only novelty in the algorithm design where the authors introduced three
> >  decay schemes and the PDE solver (the latter of which has also been widely studied by PDE people). The overall significance of this work is therefore limited.
> >
>
>  Do you refer to PINNs when you mention PDE solver? PINNs that we use to solve PDEs was introduced relatively recently. PINNs has not been studied enough yet and training PINNs is not easy.  Moreover, PINNs was mostly studied in the context of PDE equations related to physical systems, where classical solutions exist and there is no need to consider viscosity solutions in general. Thus, introducing viscosity into PINNs framework is important and novel at the same time. Let us remark that this work has another significance as it is bridging two very different communities: Scientific Machine Learning and Optimal control, which is not an easy task and indeed there is not much work done in between those fields yet.
>
>
>
> > If the authors want to enhance the theoretical contribution of this work, I would prefer to see how they would address the convergence in more details. Lemma 3.1 is a general result, but in the PINN setting, what conditions on the NN can guarantee convergence and how to characterize approximation error or convergence speed, etc. These are challenging theoretical questions if the authors want to go multiple steps further than just citing Lemma 3.1 as the foundation of their method.
>
>
> We agree that those are the important theoretical questions and they should be considered in the future work. However, performing this analysis is not straightforward, moreover analysing the convergence of neural networks is notoriously a challenging task, thus we leave it out of the scope of this paper for future work.

---

> > ### Author Response · Authors · 2023-11-20
> > **Answer Part II**
> >
> > > If the authors want to focus on the experimental aspect of their approach, I believe high dimensional control tasks may be the right playground for PINNs. The purpose of introducing NN in PDE community is for solving large dimensional problems. The value of PINN for optimal control problems are likely to be similar.
> >
> >
> > PINNs is indeed a promising solver that was introduced to overcome the curse of dimensionality. Moreover, it has an advantage over the classical numerical solver as PINNs is not mesh-dependent and thus once trained it can directly output a solution at any point of the domain. However, it is not yet widely applied to high-dimensional problems, with an exception of a few simple cases. Typical application of PINNs is finding the solution of some PDE equation which depends on $x, y, z$ and $t$, i.e. at most 4-dimensional problem.
> >
> >
> > There are a few reasons for that. PINNs are difficult to train, in particular due to a loss with different terms (mainly boundary condition term and PDE residual): controlling the respective weights of these terms in the loss is critical to convergence. Despite the  success for PINNS, this issue has limited the complexity of problems addressed through PINNS, in particular high-dimensional ones. In [3] the authors analyse this effect based on the training dynamics via the Neural Tangent Kernel (NTK) and propose a control scheme (validated in the paper with 1D PDEs). Other very recent works have addressed this issue through adaptive sampling of colocation points [6 and 7], reformulation of the loss to get a single term [5], NTK adaptive eigenvalue selection [4], taking a $l_{\infty}$ norm computed through an adversarial scheme to ensure convergence stability [1] or extension of SGD to dimensionality sampling [2]. If these approaches enable to improve PINNS training and address higher dimensional problems, we can stress that for the particular case of PINNs for HJB, authors (See [1,2] in particular) experiment on HJB with linear-quadratic-Gaussian (LQG) control guaranteeing a unique solution, and thus disregarding the difficulty of having a training scheme converging toward viscosity solutions. In this paper we take a more general stance and address this point with the $\epsilon$-schedulers. Experiments are indeed so far limited to low dimension experiments. Future work, as mentioned in the conclusion, will investigate how our algorithm could benefit from these recent approaches to improve PINNS training.
> >
> > Progress could also be made on the side of the neural architecture, that is often simple for PINNS. We made attempts with different activation functions like sine functions [8], but without experiencing any significant gain.
> >
> > Overall, it is our goal to be able to target high-dimensional problems. However, the afore-mentioned existing limitations of PINNs should be addressed first, that may involve a sequence of significant improvements in the way PINNs are trained, thus it is left for the future work.
> >
> > 1. Is $L^2$ Physics Informed Loss Always Suitable for Training Physics Informed Neural Network? https://proceedings.neurips.cc/paper_files/paper/2022/hash/374050dc3f211267bd6bf0ea24eae184-Abstract-Conference.html
> > 2. Tackling the Curse of Dimensionality with Physics-Informed Neural Networks - arXiv:2307.12306
> > 3. WHEN AND WHY PINNS FAIL TO TRAIN: A NEURAL TANGENT KERNEL PERSPECTIVE. http://arxiv.org/abs/2007.14527
> > 4. PINNACLE: PINN ADAPTIVE COLLOCATION AND EXPERIMENTAL POINTS SELECTION. https://openreview.net/pdf?id=GzNaCp6Vcg
> > 5. About optimal loss function for training physics-informed neural networks under respecting causality. http://arXiv.org/abs/2304.02282
> > 6. Mitigating Propagation Failures in Physics-informed Neural Networks using Retain-Resample-Release (R3) Sampling. http://arxiv.org/abs/2207.02338
> > 7. DeepXDE: A deep learning library for solving differential equations. https://arxiv.org/abs/1907.04502
> > 8. Implicit neural representations with periodic activation functions. https://arxiv.org/abs/2006.09661

---

> > > ### Author Response · Authors · 2023-11-20
> > > **Answer Part III**
> > >
> > > > Otherwise, with known dynamics and easily observable rewards (as in this work), it is relatively easy to come up with efficient functional forms to approximate value functions without PINNs.
> > >
> > > We are not sure what the reviewer means under "easily observable rewards" and "efficient functional forms". Does it imply that if the dynamics and reward functions are known then it should be easy to learn a neural network that approximates well the value function? If that is what implied, then we argue that in case of continuous-time reinforcement learning or optimal control it is not the case. In case of linear dynamics there are indeed more efficient algorithms developed primarily for LQR. In case of non-linear dynamics there are some numerical schemes that depend on meshes and thus have a limited scalability, similarly to FD or FEM. The NN-based approximations either work for some specific cases of optimal control problems (Darbon et al, 2023) or rely on PINNs (Lutter et al, 2020). Note that in DTRL,  knowing the transition (dynamics) and reward functions does not explicitly decrease the difficulty of finding the good functional representation for the value function (see Sutton & Barto, 2018).

---

> > > > ### Comment · Area_Chair_ULWJ · 2023-11-22
> > > > **Please respond to the author reply**
> > > >
> > > > Dear reviewer, please do respond to the author reply and let them know if this has answered your questions/concerns.

---

### Official Review · Reviewer_2hDe · 2023-10-30

**Soundness:** 3 good
**Presentation:** 3 good
**Contribution:** 3 good
**Rating:** 5
**Confidence:** 3

**Summary:**

In this paper, the authors proposed and testes a neural network based method for solving a deterministic Hamilton-Jacobi-Bellman (HJB) equation. The method utilizes the stability of viscosity sub(super)solution of HJB equation, and solves a sequence of PDEs to approximate the solution to HJB. Each PDE is solved by PINN method. Numerical experiments show that the proposed method outperforms PPO and A2C, but not the dynamic programming when it can be applied.

**Strengths:**

A new method for solving deterministic HJB, showing success in limited numerical experiments.

**Weaknesses:**

Generally, there lacks theoretical study for numerical solutions to a PDE. To ensure the necessary convergence, the conditions of Lemma 3.1 have to be verified, most importantly $W^\epsilon$ converges to $W$ uniformly. What presented in the subsection of $\epsilon$-schedulers is not adequate for this purpose.

There are also several families of numerical methods for solving HJB, not limited to the ones that the authors listed (FD,FEM) and can be easily found in literature(such as level set , the method proposed here need to be compared to them for effective and accuracy.

The numerical experiments are limited to a few small scale problems, some large scale, high dimensional examples can certainly improve the quality of the paper.

**Questions:**

Several key parameters given between equation (12) and (13) are not presented precisely, for example how to pick $k_\epsilon$ exactly for a given problem and the definition of $N_S$.

Could numerical methods for HJB be more thoroughly surveyed and compared? See e.g. Falcone, M., Ferretti, R.: Numerical methods for Hamilton-Jacobi type equations. Handb. Numer. Anal., Elsevier/North-Holland, Amsterdam,17, 603–626 (2016).

---

> ### Author Response · Authors · 2023-11-20
> **Answer. Part I**
>
> We thank the reviewer for carefully reading our paper.
> We especially thank for additional related works pointed out by the reviewer, we will add them in our related works part. About comparisons to level-set methods, as far as we are familiar with those methods, similarly to FD or FEM methods they rely on discretization of the domain $O$, which means also limited scalability of the method. See our argument in Section 4.1 in the paper. Moreover, they seem to be more suitable for finite-horizon problems, where there exists a terminal condition that $V(x, t)$ should satisfy, like $V(x, T) = \Psi (x)$ with $T$ being a terminal time, while we consider infinite-horizon problems.
>
> >Generally, there lacks theoretical study for numerical solutions to a PDE. To ensure the necessary convergence, the conditions of Lemma 3.1 have to be verified, most importantly converges to uniformly. What presented in the subsection of 𝜖-schedulers is not adequate for this purpose.
>
> We apologize for the confusion related to $\epsilon$-scheduling and uniform convergence. First of all, let us clarify that we should distinguish between the uniform convergence of equations and the uniform convergence of solutions and $\epsilon$-scheduling helps only uniform convergence of equations. Furthermore, our intention was not to say that $\epsilon$-scheduling can guarantee uniform convergence. The goal of $\epsilon$-scheduling is rather to encourage the uniform convergence. While a non-adaptive scheduler does not imply the uniform convergence of Eq.7 to Eq.4, the adaptive scheduler is designed in such a way to ensure that $\delta(\epsilon, \theta)$ converges to zero. Indeed each next $\epsilon_{n+1}$ should be small enough to ensure that $\delta(\epsilon_{n+1}, \theta_n) < \delta(\epsilon_{n}, \theta_{n-1})$. $\delta(\epsilon, \theta)$ is computed using a large number of samples from the domain $O$, samples being distributed uniformly in $O$ (related to this,  we have noticed a typo with the formula of $\delta(\epsilon, \theta)$, where instead of $N_S$ should be $N_D$). Thus, $\delta(\epsilon, \theta)$ can be used as a good proxy to understand if RHS converges to 0. We will add a better intuition behind $\delta(\epsilon, \theta)$ in the paper.
>
> As for uniform convergence of solutions, it is very hard to analyse the convergence of neural networks in terms of uniform convergence. We encourage it by introducing the regularization loss, see Eq.10. Together with uniform sampling, it helps the solution to stay close to the previously found solution for all points of the domain $O$.

---

> > ### Author Response · Authors · 2023-11-20
> > **Answer. Part II**
> >
> > >The numerical experiments are limited to a few small scale problems, some large scale, high dimensional examples can certainly improve the quality of the paper.
> >
> > We agree with the reviewer that adjusting this framework to higher-dimensional tasks will be an improvement. We definitely plan to do it in the future works. However, as we mentioned in the paper, it is not a straightforward due to a number of reasons.
> >
> >
> >
> > PINNs are  difficult to train, in particular due to a loss with different terms (mainly boundary condition term and PDE residual): controling the respective weights of these terms in the loss is critical to convergence. Despite the  success for PINNS, this issue has limited the complexity of problems addressed through PINNS, in particular high-dimensional ones. In [3] the authors analyse this effect based on the training dynamics via the Neural Tangent Kernel (NTK) and propose a control scheme (validated in the paper with 1D PDEs). Other very recent works have addressed this issue through adaptive sampling of colocation points [6 and 7], reformulation of the loss to get a single term [5], NTK adaptive eigenvalue selection [4], taking a $l_{\infty}$ norm computed through an adversarial scheme to ensure convergence stability [1] or extension of SGD to dimensionality sampling [2]. If these approaches enable to improve PINNS training and address higher dimensional problems, we can stress that for the particular case of PINNs for HJB, authors (See [1,2] in particular) experiment on HJB with linear-quadratic-Gaussian (LQG) control guaranteeing a unique solution, and thus disregarding the difficulty of having a training scheme converging toward viscosity solutions. In this paper we take a more general stance and address this point with the $\epsilon$-schedulers. So far, experiments are indeed limited to low dimension experiments. Future work, as mentioned in the conclusion, will investigate how our algorithm could benefit from these recent approaches to improve PINNS training without "breaking viscosity".
> >
> > Progress could also be made on the side of the neural architecture, that is often simple for PINNS. We made attempts with different activation functions like sine functions [8], but without experiencing any significant gain.
> >
> >
> > 1. Is $L^2$ Physics Informed Loss Always Suitable for Training Physics Informed Neural Network? https://proceedings.neurips.cc/paper_files/paper/2022/hash/374050dc3f211267bd6bf0ea24eae184-Abstract-Conference.html
> > 2. Tackling the Curse of Dimensionality with Physics-Informed Neural Networks - arXiv:2307.12306
> > 3. WHEN AND WHY PINNS FAIL TO TRAIN: A NEURAL TANGENT KERNEL PERSPECTIVE. http://arxiv.org/abs/2007.14527
> > 4. PINNACLE: PINN ADAPTIVE COLLOCATION AND EXPERIMENTAL POINTS SELECTION. https://openreview.net/pdf?id=GzNaCp6Vcg
> > 5. About optimal loss function for training physics-informed neural networks under respecting causality. http://arXiv.org/abs/2304.02282
> > 6. Mitigating Propagation Failures in Physics-informed Neural Networks using Retain-Resample-Release (R3) Sampling. http://arxiv.org/abs/2207.02338
> > 7. DeepXDE: A deep learning library for solving differential equations. https://arxiv.org/abs/1907.04502
> > 8. Implicit neural representations with periodic activation functions. https://arxiv.org/abs/2006.09661

---

> > > ### Comment · Area_Chair_ULWJ · 2023-11-22
> > > **Please respond to the author reply**
> > >
> > > Dear reviewer, please do respond to the author reply and let them know if this has answered your questions/concerns.

---

### Official Review · Reviewer_S2st · 2023-11-03

**Soundness:** 4 excellent
**Presentation:** 3 good
**Contribution:** 4 excellent
**Rating:** 8
**Confidence:** 4

**Summary:**

This paper presents a new way of solving Hamilton-Jacobi-Bellman (HJB) equation using the framework of Physics Informed Neural Network (PINN). To find the viscous solution, which is the optimal value function, this paper proposes an iterative algorithm to gradually decrease the epsilon for approximation.
The proposed algorithm works well in the pendulum task, but a problem in applying the method for higher dimension systems was identified.

**Strengths:**

This paper nicely reviews the concept of the viscous solution of HJB equation and presents a novel way of obtaining the solution by PINN, which utilized the automatic differentiation capability of recent deep learning tools.

**Weaknesses:**

Although the title is "reinforcement learning," this method requires an analytic model of the system dynamics and the cost/reward function to apply PINN, and the states are sampled uniformly randomly in the state space. It may be better called optimal control rather than reinforcement learning, which usually assumes that the agent explore the environment by its own policy without explicit prior knowledge of the environment.

**Questions:**

In Figure 1, why are the solutions have a dip at the origin, where the value function should be maximum. Is there any issue with dealing with the terminal cost/reward?
For people new to PINN, isn't it better to include the network architecture diagram with inputs, outputs, and how derivatives are combined for the objective function?

---

> ### Author Response · Authors · 2023-11-20
> **Answer to Reviewer S2st**
>
> Thank you for carefully reading the paper and highly evaluating its merit!
>
>
> > For people new to PINN, isn't it better to include the network architecture diagram with inputs, outputs, and how derivatives are combined for the objective function?
>
> Thank you for pointing it out, we will add a diagram to explain this into appendix!
>
> > It may be better called optimal control rather than reinforcement learning
>
> We acknowledge that the problem considered in this paper indeed fits well into optimal control definition. At the same time Model-Based Continuous-Time Reinforcement Learning in its definition is very closely related to optimal control and thus if the model is learned our approach is also valid for CTRL. We indeed mention that in our introduction:
>
>  "Thus, this work can be interesting for the optimal control community as we show how to use the neural networks to get the viscosity solutions and for the reinforcement learning community as our work can be used as a basis for Model Based Continuous Time Reinforcement Learning."
>
> Even though, right now we consider the case of known dynamics, it is our goal to consider the case of unknown dynamics as well. Thus, we chose Continuous-Time Reinforcement Learning as the name.
>
> >In Figure 1, why are the solutions have a dip at the origin, where the value function should be maximum. Is there any issue with dealing with the terminal cost/reward?
>
> It is an interesting question! One of the explanation (provided in the paper) is that the architecture that we use depends on smooth activation functions and has difficulty to approximate non-smooth areas. Another explanation, which concerns this zone in particular, is that only the actions that are deemed optimal by the neural network contribute to the loss computation, thus, if a chosen action is not optimal then it can introduce  error into the loss computation. This area is also the area where the best action is less distinguishable from suboptimal actions (the values in Eq.5 under argsup expression are very close to one another). A wrong choice can have a significant impact on which information is propagated during the backward propagation, e.g. different actions may have either $f(x,u) > 0$ or $f(x,u) < 0$, thus affecting the sign in front of $\nabla_x W(x)$ in $H(x, W(x), \nabla_x W(x))$. This can be observed on the image available at this link: https://anonymous.4open.science/r/review_iclr-D376/value_map_eps_%201.0seed_%201.pdf, where the action gap plot (the third one from the left) shows that areas with the lowest action gaps are the ones for which it is harder to identify the best action.

---

> > ### Comment · Area_Chair_ULWJ · 2023-11-22
> > **Please respond to the author reply**
> >
> > Dear reviewer, please do respond to the author reply and let them know if this has answered your questions/concerns.

---

### Author Response · Authors · 2023-11-22
**General comment. Part I**

We thank all the reviewers for carefully reading the paper and leaving some interesting remarks that will allow us to improve the paper.
To sum up, our paper offers a novel approach of how to make Physics Informed Neural Networks (PINNs), which is the emerging tool for solving PDE equations, find the viscosity solution of the deterministic Hamilton-Jacobi-Bellman equation that is used in optimal control and continuous-time reinforcement learning. For that we use a property of viscosity solutions, known as stability or vanishing viscosity lemma and we provide a few tricks that allow PINNs to find viscosity solutions, such as $\epsilon$-schedulers and regularization loss and we provide experimental results on a few classical control problems.

To the best our knowledge, this is the first work that integrates viscosity into PINNs. We consider the general formulation of Hamilton-Jacobi-Bellman equation (HJB) in an infinite horizon setting without assuming the shape of dynamics or reward functions. For this formulation, there may exist multiple solutions of the HJB equation, but only one viscosity solution that corresponds to the value function. Prior work on how to find viscosity solutions of the general HJB equation uses the numerical schemes such as Finite Difference and Finite Element Method that are mesh-dependent and thus limited in scalability. Other prior works that use NNs to solve HJB either do not consider viscosity (Lutter et al, 2020) or assume specific shape of dynamics or rewards (Darbon et al, 2023). We believe our work paves the way for a lot of other promising research in between optimal control, continuous-time reinforcement learning and scientific machine learning fields, from both theoretical and practical standpoints.

Overall, reviewers praised our work for proposing an interesting new method for solving deterministic HJB equation in viscosity sense using PINNs. They also appreciated us providing a clear and accessible introduction to the optimal control and viscosity solution concepts. Reviewer DLUu also valued our discussion on the existing limitations and challenges that need to be addressed in future work.

---

> ### Author Response · Authors · 2023-11-22
> **General comment. Part II**
>
> Further, we would like to address some common concerns. First, a few reviewers found that it was not clear how $\epsilon$-schedulers compare between each other and are connected to uniform convergence mentioned in Lemma 3.1. First of all, we would like to clarify that in order to satisfy conditions of Lemma 3.1 two types of uniform convergence are necessary: uniform convergence of equations (from Eq.7 to Eq.4) and uniform convergence of solutions ($W^{\epsilon} \to W$). While $\epsilon$-schedulers have only secondary effect on uniform convergence of solutions, they have a direct impact on the uniform convergence of equations. The non-adaptive scheduler is a simple way that performs regular $\epsilon$ updates at a frequency $N_{u}$, and $\epsilon$ goes to 0 with the speed $k_{\epsilon}$. On the one hand, this approach is simple and allows to control how often $\epsilon$ is updated. On the other hand, it does not depend on the value of RHS of Eq. 7, and if $k_{\epsilon}$ and $N_u$ are not chosen carefully then uniform convergence of Eq.7 to Eq.4 is not guaranteed. The adaptive scheduler does the updates according to the value of $\delta(\epsilon, \theta)$ that is a good proxy for RHS of Eq. 7. It is easy to show that $\delta(\epsilon_{n+1}, \theta_n) < \delta(\epsilon_{n}, \theta_{n-1})$, thus $\delta(\epsilon, \theta) \to 0$ when $\epsilon \to 0$. Therefore, the adaptive scheduler has more control over the uniform convergence of the equations, which is especially useful for smaller $\epsilon$. The hybrid is the combination of both schedulers using the non-adaptive at the beginning of the training for larger $\epsilon$, profiting from more regular updates, and the adaptive for smaller $\epsilon$ to have more guarantees for uniform convergence. According to our experiments, hybrid is the most robust to the choice of hyperparameters.
>
> Another common concern is the absence of high-dimensional problems in our experimentation. Let us note that we have chosen  problems of  similar complexity and dimensionality as the ones considered in Lutter et al. (2020). The latter work also uses PINNs for HJB, but without viscosity considerations. While applying the method to more high-dimensional problems is very welcome, we argue that research on PINNs has not advanced enough to solve high-dimensional problems except some special cases. There are several reasons behind it. Firstly, they are difficult to train as one should find appropriate mixing coefficients for PDE loss and boundary losses. Secondly, most PINNs methods rely on uniform sampling, which does not cover enough the difficult zones, the problem that we also observed for our cases. The adaptive sampling schemes start to appear, but still are mostly tested on the low dimensional use cases and not suited for viscosity as they may break the uniform convergence of solutions. Finally, the choice of neural network architecture can be improved to process better non-smooth areas (see more details in replies to Reviewers N25p and 2hDe).
>
> Finally, let us address the main concern of the reviewer N25p. It states that without theoretical analysis of uniform convergence or experiments on high-dimensional use cases, our work is not significant enough. We agree that both suggested improvements by Reviewer N25p are important and should be done, but we argue that those contributions can be left for future work. Indeed, the analysis of uniform convergence of NNs (NNs represent solutions) will be interesting, but it is a notoriously difficult problem. Going to high-dimensional use-cases is not yet straightforward as it is mentioned in the previous paragraph. Even without those potential improvements, we believe our work has a significant impact as it connects the popular PINN-based PDE solver that is not constrained by the dimensionality of the problem, with some core theoretical results known in optimal control (viscosity solutions and its properties). Despite the existing limitations, we think our work is pushing forward the state-of-the-art on this difficult problem and set the ground for others to advance this research topic at the crossroad between scientific machine learning, optimal control and reinforcement learning.

---

> > ### Comment · Reviewer_2hDe · 2023-11-23
> > **uniform convergence**
> >
> > Thanks to the authors for the detailed response. I am not sure that I understand the issue with respect to the concept of " uniform convergence of the equations". This is not not a common concept, and is not properly defined either in the paper or in the response. In addition, in the paragraph below equation 7, it was stated that "if $W^\epsilon$ converges uniformly to W(x), then W(x) is a viscosity solution of the original HJB equation (Equation 4)." This seems to suggest that the uniform convergence of the solutions is something one really should care about. How is this related to your statement in the response "uniform convergence of equations (from Eq.7 to Eq.4) "?

---

> > > ### Author Response · Authors · 2023-11-23
> > > **clarifying uniform convergence**
> > >
> > > **About the definition of uniform convergence of equations.** Using Lemma 3.1, the uniform convergence of Eq.7 to Eq.4 can be checked by verifying if $\epsilon \text{Tr} \nabla^2_{xx} W^{\epsilon}(x) \to 0$ uniformly as $\epsilon \to 0$, i.e. $\max_{x \in O} |\epsilon \text{Tr} \nabla^2_{xx} W^{\epsilon}(x)| \to 0$ as $\epsilon \to 0$. Note that $\epsilon$ affects not only the coefficient in RHS of Eq.7 but also $W^{\epsilon}(x)$ implicitly.
> > >
> > > **About uniform convergence of equations vs uniform convergence of solutions.** We argue that both are important. Uniform convergence of equations is necessary if we want to make sure that the final $W(x)$ is solving the original HJB equation. Indeed, in our experiments we observed that by not choosing proper hyperparameters of $\epsilon$-schedulers, $W^{\epsilon_{n+1}}(x)$ could have been arbitrarily close to $W^{\epsilon_{n}}(x)$, but not to the value function $V(x)$, while $\epsilon_{n+1} \text{Tr} \nabla^2_{xx} W^{\epsilon_{n+1}}(x)$ is not decreasing in this case, implying that $W^{\epsilon_{n+1}}(x)$ converged to the solution of some other equation. Uniform convergence of solutions is very important if we want to "propagate" viscosity from large $\epsilon$ to small $\epsilon$, but it is harder to control than the convergence of equations, as we cannot know in advance the final solution $V(x)$ and thus monitor how close we are to achieving it. To encourage the uniform convergence of solutions, we apply regularization loss (see Eq. 10), which is a trick commonly used in transfer learning. At the same time, as $\epsilon$ also affects $W^{\epsilon}(x)$ implicitly, then uniform convergence of equations can have a secondary effect on uniform convergence of solutions. Empirically, if Eq.7 does not change drastically between $\epsilon_{n+1}$ and $\epsilon_n$ then we can expect that their corresponding solutions also do not change drastically.
> > >
> > > We hope that this answer makes it clearer.

---

### Meta-Review · Area_Chair_ULWJ · 2023-12-05

**Metareview:**

(a)This paper is proposing a technique for doing continuous time RL by solving for the viscosity solution of the HJB equation. The solve the non-uniqueness issue of the HJB equation solutions. They then argue that standard methods involve some form of discretization which suffers curse of dimensionality, and propose physics informed neural nets as a solution. In particular, they show an epsilon scheduling technique allows these techniques to converge to the HJB solution.

(b) The insights provided in the paper to the best of my knowledge are novel and interesting, and there is some empirical validation on classical control tasks.

 (c) The paper falls in an awkward place in between theory and practice. The method is neither fully theoretically justified nor is it very practical in it's current instantiation. While the authors have discussed how this can be improved in the rebuttal, this still does not provide a clear path towards usability. This makes it hard to accept in it's current form.

(d) Either more theoretical justifications of why PINNs solve the problem till convergence without suffering curse of dimensionality, or more empirical experiments on more complex control domains.

**Justification For Why Not Higher Score:**

The paper falls in an awkward place in between theory and practice. The method is neither fully theoretically justified nor is it very practical in it's current instantiation. While the authors have discussed how this can be improved in the rebuttal, this still does not provide a clear path towards usability. This makes it hard to accept in it's current form.

**Justification For Why Not Lower Score:**

N/A

---

### Decision · Program_Chairs · 2024-01-16

Reject